# Understanding LLM Evaluator Behavior: A Structured Multi-Evaluator Framework for Merchant Risk Assessment

## Abstract

Large Language Models (LLMs) are increasingly used as evaluators of reasoning quality, yet their reliability and bias in *payments-risk* settings remain poorly understood. We introduce a **structured multi-evaluator framework** for assessing LLM reasoning in Merchant Category Code (MCC)-based risk assessment, combining a five-criterion domain rubric with Monte-Carlo scoring to evaluate rationale quality and evaluator stability. Five frontier LLMs generate and cross-evaluate MCC risk rationales under *attributed* and *anonymized* conditions. We develop a **consensus-deviation metric** that eliminates circularity by comparing each judge's score to the mean assigned by all *other* judges, yielding a judge-independent measure of self-evaluation bias. Results reveal substantial heterogeneity: GPT-5.1 and Claude 4.5 Sonnet exhibit negative self-evaluation bias ($-0.33$, $-0.31$), while Gemini-2.5 Pro and Grok 4 show strong positive bias ($+0.77$, $+0.71$), with bias direction persisting but attenuating 25.8% under anonymization. Validation against 26 payment industry experts reveals LLM judges score $+0.46$ points higher than human consensus, with GPT-5.1's negative bias representing closest human alignment. Empirical validation using payment network transaction data shows top-rated models achieve strongest alignment (Spearman $\rho = 0.69$–$0.77$), confirming the framework captures genuine quality. This provides a principled foundation for evaluating LLM-as-a-judge systems in payment-risk workflows and rigorous deployment protocols for operationally sensitive financial settings.

[1]Anonymous Institution, Anonymous City, Anonymous Region, Anonymous Country. Correspondence to: Anonymous Author <anon.email@domain.com>.

Preliminary work. Under review by the International Conference on Machine Learning (ICML). Do not distribute.

## 1. Introduction

Large Language Models (LLMs) are increasingly deployed not only as content generators but also as evaluators, judging the quality of text, code, and reasoning produced by other models (Chang et al., 2024; Chen et al., 2025; Kahng et al., 2024; Li et al., 2025; 2024; Tan et al., 2024; Zhang et al., 2024; Zheng et al., 2023). This LLM-as-a-judge paradigm raises foundational questions about reliability, stability, and bias—questions that are particularly salient in *payments-risk* domains, where evaluation errors can affect fraud analytics, merchant onboarding, regulatory compliance, and transaction monitoring. Merchant Category Code (MCC) risk assessment provides a demanding testbed: with more than 800 heterogeneous categories, coherent rationale generation requires integrating business-model stability, regulatory exposure, fraud typologies, return and refund behavior, and chargeback dynamics.

Despite growing interest in LLM-as-a-judge systems, no prior work provides a structured, domain-aligned evaluation of LLM reasoning in payments-risk settings. Existing benchmarks emphasize general linguistic competence or broad reasoning skills, but do not examine whether models can produce correct, complete, and operationally grounded MCC risk rationales aligned with industry practice. Moreover, prior studies rarely assess the stability of LLM-generated evaluations across repeated stochastic sampling, leaving open how consistent a given evaluator is across runs. A further limitation is the treatment of self-evaluation bias: most existing work assumes positive self-preference and uses pairwise tests that cannot measure bias magnitude, detect negative self-critique, or disentangle judge-specific bias from true quality differences (Balog et al., 2025; Dietz et al., 2025; Jiang et al., 2025; Panickssery et al., 2024; Wang et al., 2024b; Wataoka et al., 2024; Xu et al., 2024). These gaps underscore the need for a principled methodology that evaluates both the *quality* of LLM reasoning and the *behavior* of LLM evaluators in high-stakes, domain-specific contexts.

To address these challenges, we introduce a **structured multi-evaluator framework** for MCC-based merchant-risk reasoning. The framework integrates a five-criterion rubric (Accuracy, Rationale Quality, Consistency, Completeness,

Practical Applicability) with a Monte Carlo evaluation procedure in which each model performs multiple independently sampled scoring runs, yielding estimates of evaluator stability ($\mu \pm \sigma$). We evaluate five frontier LLMs—GPT-5.1, Gemini-2.5 Pro, Grok 4, Claude 4.5 Sonnet, and Perplexity Sonar—both as rationale generators and as evaluators. This design enables analysis not only of reasoning quality but also of how consistently different judges evaluate complex merchant categories under stochastic variability.

A central contribution of this work is a **consensus-deviation metric** that provides a principled method for quantifying both self-evaluation and cross-model bias. The metric eliminates circularity by comparing each judge's score only to the mean assigned by all *other* judges, ensuring that the reference standard remains independent of the judge being evaluated. This makes it possible to isolate judge-specific tendencies, measure bias magnitude robustly, and detect both positive (self-promoting) and negative (self-critical) forms of self-evaluation. Using this metric, we conduct cross-evaluation experiments under two conditions—*attributed* (source model disclosed) and *anonymized* (source concealed)—allowing us to distinguish biases driven by authorship recognition from those reflecting deeper evaluative tendencies.

Our results show substantial heterogeneity in evaluator behavior. GPT-5.1 and Claude 4.5 Sonnet exhibit *negative* self-evaluation bias ($-0.33$ and $-0.31$ points), consistently scoring their own outputs below peer consensus—a behavior not identified in prior LLM self-evaluation work. In contrast, Gemini-2.5 Pro and Grok 4 display strong *positive* bias ($+0.77$ and $+0.71$), while Perplexity Sonar exhibits modest positive bias ($+0.21$). Bias direction persists under anonymization: although anonymization reduces magnitude by 25.8% on average, it does not reverse direction, indicating that evaluator tendencies reflect underlying model characteristics rather than explicit authorship cues. These findings demonstrate that evaluation behavior varies systematically across models and can differ sharply from traditional expectations of universal self-preference.

To validate these LLM-based findings, we conducted complementary human expert evaluation: 26 domain experts from the payments industry—including research scientists and experienced business partners with deep expertise in merchant risk assessment, fraud prevention, and payment operations—independently evaluated the same LLM-generated rationales using the identical evaluation rubric. This human validation reveals that LLM judges systematically assign scores averaging +0.46 points higher than human expert consensus, with models exhibiting negative bias relative to LLM peers (GPT-5.1, Claude-4.5 Sonnet) demonstrating closest alignment with human judgment. We further validate findings against four years of payment network

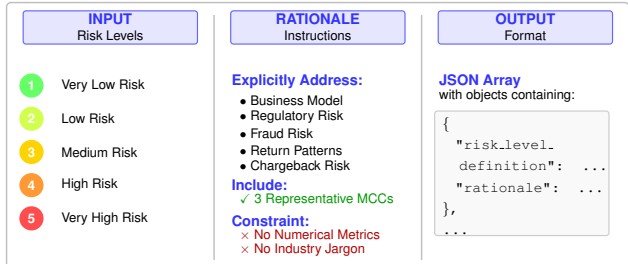

*Figure 1.* **Structure of the MCC Risk Rationale Prompt. Left**: INPUT specifies five risk levels from very low to very high risk. **Center:** RATIONALE instructions require explicit coverage of five payments-risk dimensions (Business Model Stability, Regulatory Exposure, Fraud Exposure, Return Patterns, Chargeback Activity) along with 3 representative MCCs, while prohibiting numerical metrics and industry jargon. **Right:** OUTPUT format specifies a structured JSON array containing risk level definitions and rationales. Full prompt text appears in Appendix C.

transaction data, showing that top-rated models achieve Spearman $\rho = 0.69$–$0.77$ with empirical merchant risk patterns. This triangulated validation—combining peer consensus, human expert assessment, and empirical ground truth—confirms that the evaluation framework captures genuine quality differences rather than shared model artifacts.

**Contributions.** We provide the first structured, domain-aligned evaluation of LLM reasoning in MCC-based *payments-risk* settings, introducing a Monte Carlo framework that quantifies scoring stability and a consensus-deviation metric with proven circularity prevention. Our analysis reveals the first quantitative evidence of negative self-evaluation bias in frontier LLMs, with bias direction persisting under anonymization despite 25.8% magnitude reduction. Validation through 26 domain experts and payment network transaction data demonstrates that models exhibiting conservative scoring relative to LLM peers align more closely with human judgment and empirical risk patterns.

## 2. MCC Risk Rationale Generation

Figure 1 illustrates the prompt design used to elicit structured MCC risk rationales that serve as evaluation targets in our multi-evaluator framework. This section describes the prompt structure, the five core risk dimensions, and the LLM-generated outputs analyzed in later sections.

### 2.1. Prompt Design and Risk Level Specification

Five frontier LLMs—GPT-5.1, Gemini-2.5 Pro, Grok 4, Claude-4.5 Sonnet, and Perplexity Sonar—were each instructed to act as global payment-risk experts and produce qualitative assessments across a five-level spectrum (Very Low to Very High Risk). To ensure comparability across models, the prompt requires each rationale to address five

**1 — Very Low Risk**

Merchants operate highly stable, predictable business models with minimal seasonal variation and long operational histories. Low regulatory complexity with straightforward compliance. Fraud exposure is minimal due to transparent transaction patterns. Returns are rare. Chargeback activity is virtually non-existent.

**Representative MCCs**

- 5411 — Grocery Stores
- 5912 — Drug Stores & Pharmacies
- 5541 — Service Stations

**2 — Low Risk**

Merchants exhibit stable business models with moderate predictability and occasional seasonal fluctuations that are well-understood and manageable. Regulatory exposure is routine with standard licensing. Fraud exposure is occasional and low-value. Return patterns are predictable and align with industry norms. Chargeback activity is low and manageable.

**Representative MCCs**

- 5812 — Restaurants
- 5661 — Shoe Stores
- 5943 — Stationery Stores

**3 — Medium Risk**

Merchants operate in moderately stable environments with noticeable seasonal peaks or cyclical demand patterns. Regulatory exposure involves multiple compliance layers and periodic audits. Fraud exposure is moderate with recurring attempts. Return patterns are significant and often seasonal or trend-driven. Chargeback activity requires dedicated resources to manage.

**Representative MCCs**

- 5651 — Family Clothing Stores
- 5732 — Electronics Stores
- 5945 — Hobby, Toy & Game Stores

**4 — High Risk**

Merchants face elevated business model instability due to rapid market changes, high competition, or dependency on discretionary consumer spending. Regulatory exposure is substantial with complex compliance frameworks. Fraud exposure is frequent and involves sophisticated schemes. Return patterns are high-volume and unpredictable. Chargeback activity is endemic and requires intensive management.

**Representative MCCs**

- 5816 — Digital Goods: Games
- 5967 — Inbound Telemarketing
- 7995 — Betting/Casino/Lotto

**5 — Very High Risk**

Merchants operate in highly volatile or controversial sectors prone to sudden business collapse, reputational damage, or legal challenges. Regulatory exposure is extreme with constantly evolving legal landscapes. Fraud exposure is pervasive involving organized criminal activity and identity theft. Return patterns are chaotic. Chargeback activity is at crisis levels, threatening merchant account viability.

**Representative MCCs**

- 5933 — Pawn Shops
- 7273 — Dating & Escort Services
- 5968 — Continuity/Subscription

*Figure 2.* **Example LLM-Generated MCC Risk Rationales (Claude-4.5 Sonnet).** Each rationale synthesizes all five risk dimensions and selects representative MCCs. Color gradients reflect increasing risk severity. Complete outputs for all models appear in Appendix D.

domain-relevant dimensions (Business Model Stability, Regulatory Exposure, Fraud Exposure, Return/Refund Patterns, Chargeback Activity) and select three representative MCCs from the over 800 distinct merchant categories (Mastercard, 2023; Visa Inc., 2023). To promote domain-aligned, interpretable reasoning, the prompt restricts numerical claims and discourages specialized industry jargon, and the output must follow a structured JSON schema. The five-level structure yields concise prompts and facilitates interpretable reasoning consistent with payments-risk practice. Importantly, all LLMs had access only to a public MCC-to-Name mapping table; no transaction-level, merchant-level, or proprietary network data were provided, so all generated rationales reflect general knowledge learned during pretraining.

### 2.2. LLM-Generated Risk Rationales

Each model produces structured JSON outputs containing five rationales—one per risk level—covering all required

dimensions and corresponding representative MCCs. These outputs constitute the core artifacts evaluated for reasoning quality, cross-model agreement, bias, and stability in Section 3.

Figure 2 shows example rationales generated by Claude-4.5 Sonnet. Each card reflects the five required dimensions, a coherent shift in language intensity across risk levels, and representative MCCs aligned with industry expectations (e.g., essential services at lower risk levels and high-volatility or high-chargeback sectors at higher levels), despite models' access only to category names.

Across models, the rationales show systematic structure and domain-relevant distinctions, providing a consistent basis for the cross-model evaluation framework introduced in Section 3. Full JSON outputs for all five LLMs are provided in Appendix D.

**EVALUATION CONTEXT**

**Role: Global Payments-Risk Domain Expert**

**Target Models (5 LLMs):**
• OpenAI GPT-5.1
• Gemini 2.5 Pro
• Grok 4
• Claude 4.5 Sonnet
• Perplexity Sonar

**MONTE CARLO PROTOCOL**

**10 Independent Runs**
Each run re-scores the same fixed rationale text

**Temperature = 0.7**
Sampling variability quantifies evaluator stability

**Result:** $\mu \pm \sigma$
  $\mu$ = mean score
  $\sigma$ = consistency measure

**CRITICAL RULES**

**Rule 1:**
Do NOT alter LLM outputs
Evaluation only, no modification

**Rule 2:**
10 runs = 10 independent
scoring passes by YOU (the evaluator)

No new rationales generated
Only evaluator's judgment repeated
10 times

**SCORING RUBRIC (0–10)**

**1. Accuracy**
0-3: Incorrect
4-6: Missing $\geq 2$ drivers
7-8: Mostly correct
9-10: Fully aligned

**2. Rationale Quality**
0-3: Unclear
4-6: Lacks depth
7-8: Clear & structured
9-10: Polished

**3. Consistency**
0-3: Contradictory
4-6: Some gaps
7-8: Smooth
9-10: Precise logic

**4. Completeness**
Check 5 dims × 5 levels
Score = (Points / 25) × 10

**5. Practical Applicability**
0-3: Too vague
4-6: Some utility
7-8: Useful
9-10: Strong value

**PROCEDURE**

**A.** Run 10 Monte Carlo samples
**B.** Score each criterion (0-10)
**C.** Provide justification
**D.** Compute statistics:
  • $\mu$ (mean)
  • $\sigma$ (std deviation)
**E.** Report $\mu \pm \sigma$ per criterion
**F.** List strengths/weaknesses

**REQUIRED OUTPUT**

**1. Criterion Scores**
Format: $\mu \pm \sigma$ + Justification

**2. Final Total Score**
Format: $\mu_{\text{total}} \pm \sigma_{\text{total}}$

**3. One-Paragraph Expert Synthesis**

**4. Message Structure:**
  • Msg 1-5: Each LLM
  • Msg 6: Summary table

*Figure 3*. **Monte Carlo Evaluation Framework**. **Top**: *Evaluation Context* specifies the evaluator role and target models, *Monte Carlo Protocol* defines the 10-run sampling procedure at temperature 0.7 to quantify stability, and *Critical Rules* prohibit output modification and clarify that runs represent repeated evaluator judgments. **Bottom:** *Scoring Rubric* provides 0–10 scales for five criteria (Accuracy, Rationale Quality, Consistency, Completeness, Practical Applicability), *Procedure* outlines the six-step evaluation workflow, and *Required Output* specifies the structured reporting format with $\mu \pm \sigma$ scores and expert synthesis. Full prompt text appears in Appendix E.

## 3. LLM-as-Judge Evaluation with Rubric and Monte Carlo Stability

This section presents the framework used to evaluate the quality and consistency of the MCC risk rationales generated in Section 2. The same five frontier LLMs act as evaluators, scoring one another's rationales using a structured rubric and a Monte Carlo protocol to measure evaluator stability.

### 3.1. Monte Carlo Evaluation Framework

Figure 3 summarizes the evaluation setup. Each LLM is cast as a "Global Payments-Risk Domain Expert" and evaluates all rationales, including its own. The evaluation protocol has three components: (1) a role-establishing context, (2) a five-criterion scoring rubric, and (3) a Monte Carlo sampling procedure to quantify evaluator stability.

**Monte Carlo Protocol.** For each judge–target pair, we perform 10 independent scoring runs at Temperature=0.7. Critically, the evaluated rationale remains fixed across all runs; only the evaluator's stochastic reasoning varies. Thus, $\mu$ and $\sigma$ reflect *evaluator* uncertainty rather than content

variation (detailed algorithm in Appendix F).

**Scoring Rubric.** Evaluators score each rationale on five dimensions: **Accuracy** (alignment with known MCC risk behavior), **Rationale Quality** (clarity, structure, domain professionalism), **Consistency** (coherent risk progression), **Completeness** (coverage of all required dimensions), and **Practical Applicability** (usefulness for decision making).

### 3.2. Attributed vs. Anonymized Evaluation Conditions

To separate identity-driven effects from deeper evaluator tendencies, scores are collected under two conditions: **Attributed** (judges see full model identities) and **Anonymized** (identities replaced with "Expert 1–5" labels). This dual-condition design isolates the influence of authorship recognition.

### 3.3. Cross-Evaluation Score Matrices

The full Monte Carlo evaluation yields two $5 \times 5$ matrices—one for attributed scoring and one for anonymized scoring—shown in Figure 4. Each cell reports a judge's stabilized mean score ($\mu$) and consistency estimate ($\sigma$). Color

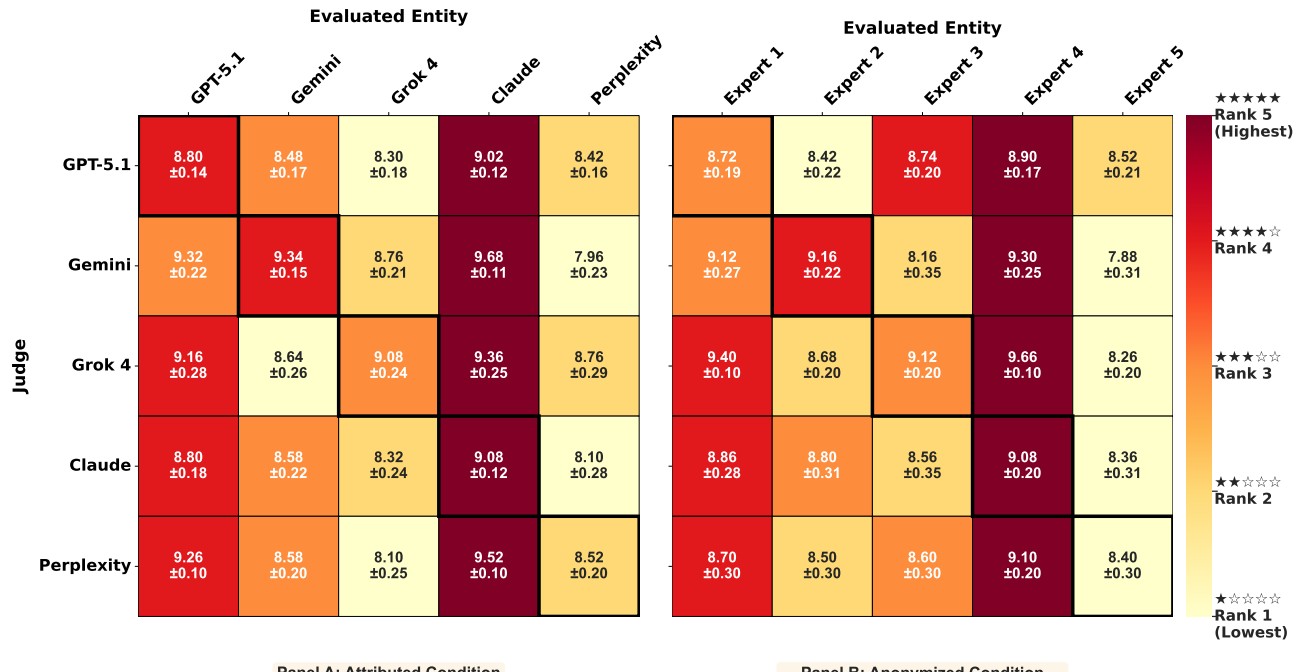

*Figure 4.* **Cross-Evaluation Score Matrices.** Five LLMs evaluate all MCC rationales under (**A**) attributed and (**B**) anonymized conditions. Each cell reports mean ± standard deviation from 10 Monte Carlo runs. Color shading reflects relative rankings. Diagonal entries represent self-evaluations.

shading encodes each judge's *relative* ordering of the five targets, enabling comparison of ranking behavior independent of absolute calibration differences. Detailed criterion-level numerical values corresponding to these matrices are provided in Appendix G. Three observations follow directly from these matrices.

**(1) Self-scoring varies substantially across models.** Diagonal entries show that models differ in how favorably they evaluate their own rationales (e.g., Gemini-2.5 Pro: 9.34; Grok 4 and Claude-4.5 Sonnet: 9.08; GPT-5.1: 8.80; Perplexity Sonar: 8.52). These differences do not yet imply positive or negative *bias*—they simply show that raw self-scores are not uniform across evaluators.

**(2) Anonymization shifts score magnitudes but preserves ranking patterns.** Comparing Panels A and B indicates that anonymization slightly increases or decreases scores depending on the judge–target pair (e.g., Gemini-2.5 Pro: 9.34 → 9.16; GPT-5.1: 8.80 → 8.72). While these differences are easily visible, the matrices alone cannot determine whether the changes constitute positive or negative bias. That requires a judge-independent baseline, introduced in Section 4.

**(3) Peer judges show clear areas of agreement.** Column patterns reveal strong consensus: Claude-4.5 Sonnet receives uniformly high peer evaluations across both conditions; GPT-5.1 also receives consistently strong ratings; and Perplexity Sonar receives the lowest peer scores. These differences highlight shared judgments across evaluators.

Together, these observations motivate the need for a principled method to compare each model's self-assessment against the consensus of other judges. The consensus-deviation metric introduced in Section 4 formalizes this comparison and enables rigorous quantification of evaluation bias.

### 3.4. Human Expert Validation

Twenty-six payment industry professionals independently evaluated all five LLM-generated rationales using the identical five-criterion rubric with blinded model identities. Human consensus (Figure 5) shows Claude-4.5 Sonnet (8.88 ± 0.76) and GPT-5.1 (8.81 ± 0.67) ranked highest, followed by Gemini-2.5 Pro (8.20), Grok-4 (8.05), and Perplexity Sonar (7.73), largely aligning with LLM peer consensus. Detailed analysis in Section 4.4 shows models exhibiting negative bias relative to LLM peers align most closely with human evaluation standards.

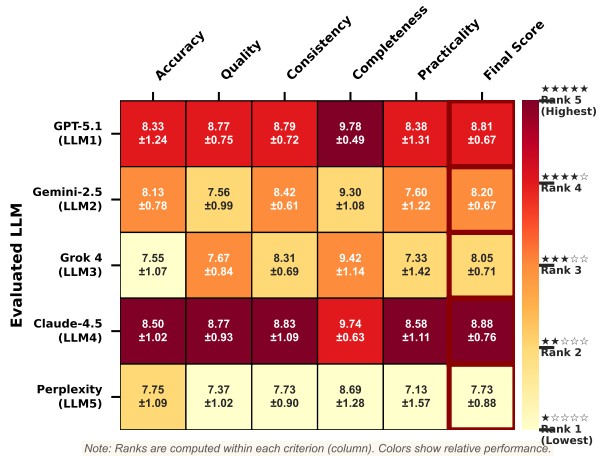

*Figure 5.* **Human Expert Consensus on LLM Risk-Assessment Quality.** Twenty-six payment-industry experts evaluated five LLM-generated rationales across five criteria. Cells show mean $\pm$ standard deviation with shading indicating relative rankings. Claude-4.5 and GPT-5.1 receive highest scores.

# 4. Bias Metrics: Mathematical Formulation and Empirical Characterization

Section 3 introduced raw scores from the Monte Carlo LLM-as-judge framework. We now formalize conversion to a judge-independent measure of deviation from peer consensus. This section defines the *consensus-deviation metric*, establishes theoretical guarantees, and characterizes systematic evaluation tendencies across frontier LLMs.

## 4.1. Definition of the Consensus-Deviation Metric

The consensus-deviation metric measures a judge's score deviation from consensus formed by all *other* judges. Excluding the focal judge ensures independence and eliminates circularity.

### 4.1.1. NOTATION

Let $n$ be the number of judges and $m$ the number of evaluated entities. For each judge $i \in \{1, \ldots, n\}$ and entity $j \in \{1, \ldots, m\}$, $\text{Score}_{\text{judge}=i}(\text{LLM} = j)$ denotes the attributed score and $\text{Score}_{\text{judge}=i}(\text{Expert} = j)$ the anonymized score, each computed as the Monte Carlo mean from 10 runs. Consensus for entity $j$ excludes judge $i$, ensuring independence.

### 4.1.2. ATTRIBUTED BIAS

When identities are visible, deviation from consensus is:

$$\text{Bias}_A(i, j) = \text{Score}_{\text{judge}=i}(\text{LLM} = j)$$
$$- \text{MeanScore}_{k \neq i}(\text{LLM} = j).$$

The diagonal case $i = j$ corresponds to self-evaluation bias.

### 4.1.3. ANONYMIZED BIAS

Under anonymization, identity labels are replaced using a fixed mapping: Expert $j$ = LLM $j$.

The deviation is:

$$\text{Bias}_B(i, j) = \text{Score}_{\text{judge}=i}(\text{Expert} = j)$$
$$- \text{MeanScore}_{k \neq i}(\text{Expert} = j).$$

Here, $\text{Bias}_B(i, i)$ captures intrinsic self-evaluation tendencies independent of authorship disclosure.

## 4.2. Theoretical Properties and Guarantees

The consensus-deviation metric satisfies two key guarantees distinguishing it from naive, self-inclusive baselines (proofs in Appendix H).

### 4.2.1. PROPOSITION 1: ZERO-SUM PROPERTY ACROSS JUDGES

**Proposition 4.1.** *For any entity* $j$,

$$\sum_{i=1}^{n} Bias_A(i, j) = 0, \qquad \sum_{i=1}^{n} Bias_B(i, j) = 0.$$

**Interpretation.** The zero-sum structure ensures positive bias by some judges necessitates negative bias by others. The metric measures *relative* deviation from collective judgment, not absolute quality. If all judges assign identical scores, every bias term is zero, ensuring the metric reflects divergence from peer consensus rather than raw scoring scale.

### 4.2.2. PROPOSITION 2: SELF-EXCLUSION PREVENTS CIRCULARITY

**Proposition 4.2.** *Because consensus excludes judge* $i$,

$$\frac{\partial Bias_A(i, j)}{\partial Score_{\text{judge}=i}(LLM = j)} = 1.$$

**Interpretation.** Self-exclusion ensures judges cannot influence the baseline against which their deviation is computed (derivative = 1). This isolates *genuine evaluator bias* from scoring-scale differences—without it, judges would contaminate their own baselines, attenuating measured deviations.

### 4.2.3. CONTRAST WITH NON-EXCLUDING CONSENSUS

If consensus included the focal judge, the derivative would be $(n - 1)/n$ instead of 1, inducing a $(n - 1)/n$ contraction that systematically underestimates true deviation (e.g.,

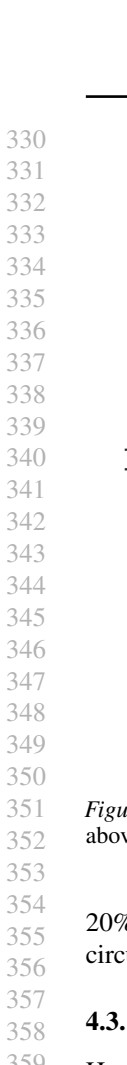

**Evaluated Entity**

**Evaluated Entity**

Panel A: Attributed Condition
(Bias = Judge Score - Peer Consensus)

Panel B: Anonymized Condition
(Bias = Judge Score - Peer Consensus)

*Figure 6.* **Bias Matrices Under Attributed (A) and Anonymized (B) Conditions.** Deviation from peer consensus (blue = below, red = above). Diagonals show self-evaluation; off-diagonals show cross-model biases.

20% attenuation for $n = 5$). Self-exclusion removes this circularity, yielding unbiased estimates.

### 4.3. Empirical Characterization of Evaluation Bias

Having established the theoretical properties, we examine frontier LLM evaluator behavior. Figure 6 shows consensus-deviation matrices aggregating Monte Carlo–stabilized scores under both attribution conditions. Each cell represents judge $i$'s deviation from peer consensus when scoring entity $j$ (blue = under-scoring, red = over-scoring). Diagonals quantify self-evaluation bias; off-diagonals reveal cross-model tendencies.

#### 4.3.1. CENTRAL FINDING: HETEROGENEOUS SELF-EVALUATION BEHAVIOR

Diagonal cells show that frontier LLMs differ markedly in self-evaluation. Two high-performing models display negative self-evaluation bias: GPT-5.1 has Bias A $(1,1)$ = –0.33 and Bias B $(1,1)$ = –0.30, while Claude-4.5 has Bias A $(4,4)$ = –0.31 and Bias B $(4,4)$ = –0.16. Both models rate their own rationales below peer consensus despite strong external scores, indicating a self-critical evaluation style rather than weaker content.

In contrast, two models show strong positive self-evaluation bias: Gemini 2.5 Pro at Bias A $(2,2)$ = +0.77 and Bias B $(2,2)$ = +0.56, and Grok-4 at Bias A $(3,3)$ = +0.71 and Bias B $(3,3)$ = +0.60. Perplexity Sonar shows milder positive bias at Bias A $(5,5)$ = +0.21 and Bias B $(5,5)$ = +0.15.

These patterns map a continuum from self-critical to self-affirming to nearly neutral, highlighting distinctive evaluative signatures across frontier models.

#### 4.3.2. STABILITY ACROSS CONDITIONS: EVIDENCE OF INTRINSIC EVALUATOR HEURISTICS

No model changes the *direction* of its self-evaluation bias when authorship labels are removed (GPT-5.1: $-0.33 \rightarrow -0.30$; Gemini-2.5 Pro: $+0.77 \rightarrow +0.56$; Grok-4: $+0.71 \rightarrow +0.60$; Claude-4.5: $-0.31 \rightarrow -0.16$; Perplexity: $+0.21 \rightarrow +0.15$). Bias magnitudes shrink by 9.1–48.4% (mean 25.8%), but *sign is preserved for all five models*. This stability indicates self-evaluation arises from *intrinsic scoring heuristics*, not explicit identity recognition.

#### 4.3.3. OFF-DIAGONAL EVIDENCE: UNIVERSAL STANDARDS VS. MODEL PREFERENCE

Off-diagonal cells show negative-bias models apply stricter standards broadly (e.g., Claude-4.5 assigns negative deviations to both itself and GPT-5.1), while positive-bias models (Gemini-2.5 Pro, Grok-4) apply more generous scoring across targets. This indicates self-bias reflects consistent evaluator style rather than isolated effects.

### 4.4. Human Expert Baseline Comparison

To determine whether LLM peer-relative patterns reflect genuine differences or shared artifacts, we compare against an independent human expert baseline. Comparing LLM

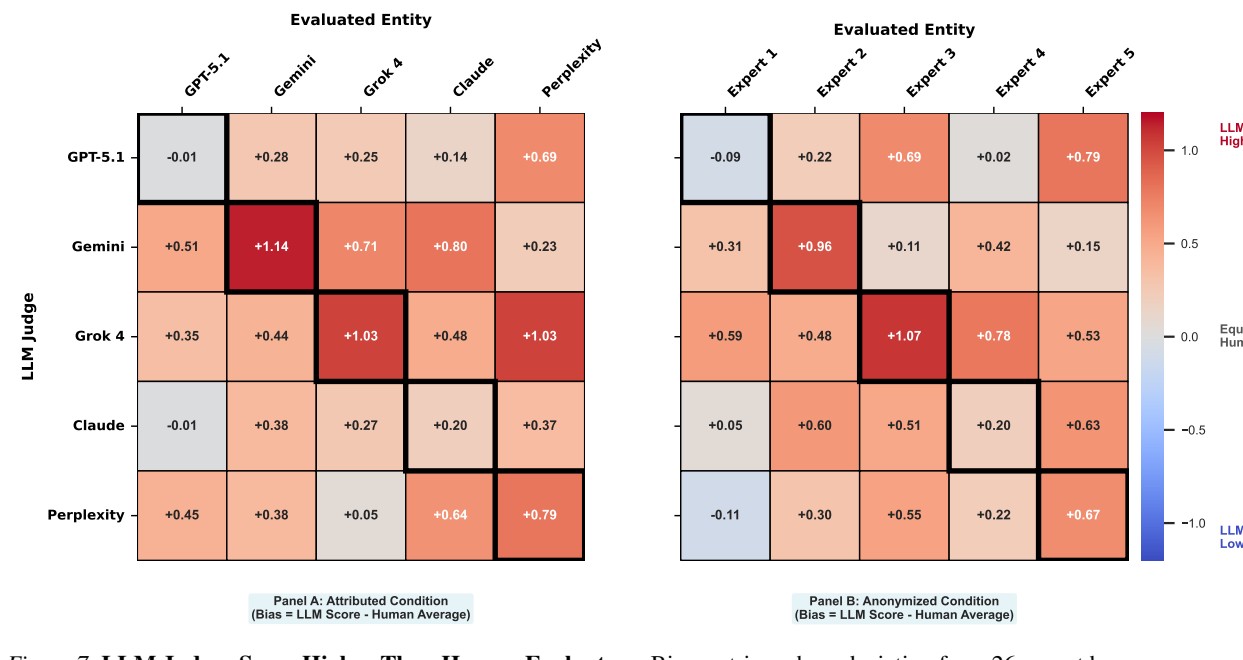

*Figure 7.* **LLM Judges Score Higher Than Human Evaluators.** Bias matrices show deviation from 26-expert human consensus (red = higher LLM scores, blue = lower). Diagonal elements (bold) reveal self-evaluation bias. **Panel A:** Mean bias $+0.46$ across 23/25 pairs when attributed. **Panel B:** Mean bias $+0.43$ when anonymized. GPT-5.1 shows near-zero self-bias ($-0.01$), while Gemini-2.5 Pro ($+1.14$) and Grok-4 ($+1.03$) show largest positive self-biases.

judges with human consensus (Figure 7) shows LLM judges assign scores on average 0.46 points higher than 26 payment-industry experts, indicating more lenient standards. Against this baseline, GPT-5.1's negative peer-relative bias ($-0.33$) emerges as *closest alignment with human judgment* ($-0.01$), while Claude-4.5's peer-relative negativity ($-0.31$) translates to mild positive bias versus humans ($+0.20$). Gemini-2.5 Pro and Grok-4 exhibit substantially stronger positive bias versus humans ($+1.14$ and $+1.03$). These findings reframe evaluator behavior: models appearing self-critical relative to LLM consensus most closely reflect human domain-expert judgment, suggesting conservative self-evaluation corresponds to realism rather than harshness.

## 5. Empirical Validation Against Payment Network Transaction Data

Sections 4.3 and 4.4 showed that LLM evaluators exhibit systematic biases relative to peer consensus and human experts. A key question remains: do these patterns reflect genuine risk-assessment skill or merely shared heuristics? We test this by validating LLM-generated risk assessments against four years of payment-network transactions spanning 800+ MCCs, focusing on the 39 MCCs highlighted in LLM rationales. Data, methods, and implications of this validation appear in Appendix I.

Four models show statistically significant alignment with

empirical risk: Claude-4.5 ($\rho = 0.77$, $p < 0.001$), Gemini-2.5 Pro ($\rho = 0.69$, $p < 0.01$), Grok-4 ($\rho = 0.61$, $p < 0.05$), and GPT-5.1 ($\rho = 0.56$, $p < 0.05$). Perplexity Sonar shows weaker non-significant correlation ($\rho = 0.49$, $p = 0.063$). Notably, Claude and Gemini—top rated by both LLM peers and human experts—also achieve strongest empirical accuracy. This three-way convergence indicates that the evaluation framework captures genuine model quality rather than shared artifacts.

Beyond aggregate correlations, examining which categories each model assigns to specific risk levels illuminates model behavior and failure modes. Figure 8 in Appendix I provides this comparison across all five models.

## 6. Related Work, Discussions, and Limitations

Related work, methodological implications, model-development and deployment considerations, and limitations and future directions are detailed in Appendix A and Appendix B.

## 7. Conclusion

We introduced a rigorous framework for evaluating LLM-as-a-judge systems in financial settings, combining a domain-aligned scoring rubric with a consensus-deviation metric that isolates self-evaluation bias while avoiding circularity. Validation through three independent

sources—peer consensus among five frontier models, assessment by 26 payment-industry experts, and alignment with payment-network transaction outcomes—demonstrates that the framework captures meaningful differences in evaluator quality rather than superficial scoring artifacts. This provides a principled and validated foundation for analyzing LLM evaluators in payment-risk workflows, showing that systematic evaluator biases can be measured, compared to human expertise, and grounded in empirical outcomes to support reliable and transparent AI-driven financial decision-making.

## 8. Impact Statement

This work advances the evaluation of Large Language Models by introducing a principled, bias-aware framework for assessing LLM-as-a-judge behavior in high-stakes financial settings. By quantifying evaluator bias, aligning judgments with human expertise, and validating assessments against real transaction outcomes, the framework supports more transparent, reliable, and accountable AI-assisted decision processes.

Potential societal impacts include improved fairness and consistency in payment-risk workflows, as well as clearer identification of where human oversight is necessary. At the same time, deploying LLM-based evaluators in financial contexts requires attention to issues of bias, robustness, and responsible use. Our analysis highlights these considerations and provides tools for monitoring evaluator behavior, helping ensure that AI-driven assessments are applied safely. Beyond these points, we are not aware of additional ethical or societal implications that require specific attention.

## Acknowledgement

[To be filled by the authors]

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

## A. Related Work

**LLM-as-a-judge systems** have become essential tools for evaluating summarization, reasoning, and safety in language models. Established methodologies include rubric-based scoring (Li et al., 2025; Liu et al., 2024; 2023; Murugadoss et al., 2025; Wu et al., 2023; Zheng et al., 2023; Zhong et al., 2024), pairwise preference models (Feng et al., 2025; Jeong et al., 2025; Panickssery et al., 2024; Zheng et al., 2023), and self-consistency protocols (Feng et al., 2025; Wang et al., 2022). While these frameworks demonstrate strong alignment with human preferences, recent work reveals systematic biases including self-enhancement, verbosity preference, position effects, and prompt-format sensitivity (Balog et al., 2025; Chen et al., 2024; Herrera-Berg et al., 2023; Li et al., 2023; Panickssery et al., 2024; Saito et al., 2023; Wang et al., 2024b; Wu et al., 2023; Ye et al., 2024). However, prior studies evaluate self-assessment in isolation rather than within multi-judge environments, leaving open whether frontier models exhibit distinct and persistent self-evaluation patterns when assessed against peer consensus.

**Validation against external baselines** remains a central challenge for LLM evaluators. While prior work compares model judgments with human preferences on general tasks (Zheng et al., 2023; Dubois et al., 2024) or explores calibration against task-specific ground truth (Qi et al., 2025), few studies provide triangulated validation combining peer consensus, domain-expert assessment, and empirical outcome data.

**Payments-risk assessment** has traditionally relied on structured features and expert-defined heuristics for fraud detection and merchant classification (Bahnsen et al., 2016; Wang et al., 2024a; Yeh et al., 2020; Zhang et al., 2021). Recent industry efforts apply foundational models to transaction embeddings and risk explanation (Dou et al., 2025; Pirmorad, 2025; Sanz-Guerrero & Arroyo, 2024; Tan et al., 2025; Yeh et al., 2025), though evaluation standards for LLM-generated rationales remain nascent.

## B. Discussions, Implications, and Limitations

Our findings reveal LLMs exhibit structured, model-specific evaluation behaviors persisting across attribution conditions. This section discusses implications for evaluation methodology, model development, and deployment, followed by limitations and future directions.

### B.1. Evaluator Diversity Versus Generation Homogeneity

Jiang et al. (Jiang et al., 2025) demonstrate that 70+ language models display notable homogeneity in generated content, forming an "artificial hivemind." Our results reveal a contrasting picture for evaluation: the bias spectrum spans more than one full point (from $-0.33$ to $+0.77$), indicating substantial heterogeneity in evaluator tendencies. While self-critical models converge—both GPT-5.1 and Claude-4.5 exhibit negative bias—this shared tendency exists within a broader landscape of divergent behaviors, highlighting that LLM generation and evaluation are governed by distinct behavioral modes.

### B.2. Metacognition, Self-Critique, and Cognitive Analogues

Negative self-evaluation bias in high-performing models, validated against domain-expert assessments, suggests model-level "self-critique" inviting comparison with human cognition. Kruger and Dunning (Kruger & Dunning, 1999) show low performers overestimate competence while high performers underestimate it—the Dunning–Kruger effect. Our findings reveal potential LLM analogues: weaker models exhibit stronger self-promotion, while GPT-5.1 and Claude-4.5 systematically under-score themselves relative to LLM peers, resembling "imposter syndrome" (Clance & Imes, 1978).

Critically, the 26-expert baseline reframes these observations: "negative bias" versus LLM peers actually represents *closer alignment with human expert judgment*. GPT-5.1 exhibits near-zero bias versus humans ($-0.01$ attributed, $-0.09$ anonymized) while its negative bias versus LLMs ($-0.33$) reflects that most LLM judges score more generously than humans. Claude-4.5 shows modest positive bias versus humans ($+0.20$) despite negative bias versus LLM peers ($-0.31$). In contrast, Gemini-2.5 Pro and Grok-4 exhibit substantial positive bias versus humans ($+1.14$ and $+1.03$), indicating systematic overestimation.

These patterns, grounded in expert validation, raise questions about evaluator bias origins. Claude-4.5's conservative scoring may reflect Anthropic's Constitutional AI (Askell et al., 2021; Bai et al., 2022a;b), which trains models to critique and revise outputs through iterative self-improvement, potentially embedding conservative tendencies aligned with human standards. Similarly, GPT-5.1's behavior may arise from RLHF processes rewarding caution and penalizing overconfidence (Mu et al.,

2024; Ouyang et al., 2022; Yuan et al., 2025). That these approaches produce evaluation behavior more consistent with expert judgment than typical LLM scoring suggests they embed realistic quality standards rather than artificial harshness.

In contrast, models with positive bias relative to both LLM peers and humans (Gemini-2.5 Pro, Grok-4) may reflect different alignment objectives prioritizing confidence, or lack explicit self-critique mechanisms. Understanding these divergent training philosophies represents a promising interpretability research direction.

While LLMs lack human metacognition phenomenologically, these structured patterns, validated across multiple baselines, indicate their evaluation heuristics embed domain-appropriate calibration rather than arbitrary noise. The evidence strengthens that negative bias versus LLM peers signals realistic assessment, and that training emphasizing self-critique produces evaluators aligned with expert judgment.

### B.3. Implications for Practice and Deployment

Observed heterogeneity carries direct consequences for benchmarking and deployment. Evaluator selection materially affects outcomes: positive-bias models may inflate scores, while negative-bias models may penalize strong reasoning. Judge disagreement reflects systematic differences in learned standards rather than noise, requiring practitioners to treat evaluators as models requiring calibration and monitoring.

In high-stakes settings (payments risk, compliance, model governance), these patterns become more consequential. Three strategies emerge: First, post-hoc calibration can normalize scores across models with different tendencies. Second, multi-judge ensembles aggregating diverse biases may yield more robust assessments (Li et al., 2023; Pal et al., 2024). Third, auditing must extend beyond output quality to evaluator tendencies, since our metric identifies model-specific behaviors independent of content.

### B.4. Limitations and Future Directions

While our study provides the first systematic quantification of self-evaluation bias in LLMs, several limitations warrant discussion.

**Limited Model Coverage.** We evaluate five models—GPT-5.1, Gemini-2.5 Pro, Grok-4, Claude-4.5, and Perplexity Sonar—representing a subset of deployed LLMs. Bias patterns may differ for open-source models (LLaMA, Mistral, DeepSeek, Qwen), smaller scales, domain-specialized models, or multilingual evaluation. Observed behaviors may characterize high-end proprietary models rather than universal tendencies.

**Static Model Snapshots.** Our analysis considers specific versions at a single point. As models update, scoring heuristics may evolve. Negative self-bias in GPT-5.1 or Claude-4.5 may reflect current tuning and could shift in later releases. Findings should be interpreted as version snapshots rather than stable properties.

**Single Task Domain.** We focus exclusively on payment risk assessment. While this provides well-structured tasks and clear criteria, self-critical or self-promotional tendencies may vary across other domains (creative writing, math, code, dialogue, QA). Understanding cross-domain generality remains important future work.

**Mechanistic Ambiguity.** Although bias persists under anonymization, we cannot fully disentangle: (1) universal evaluation standards learned during training, applied uniformly; versus (2) implicit style recognition detecting characteristic features without explicit labels. Several patterns support universal standards—e.g., Claude-4.5 assigns similar negative deviations to GPT-5.1 as to itself, and Constitutional AI teaches quality assessment independent of authorship—but controlled experiments with style transfer are needed for conclusive attribution. This ambiguity does not affect practical implications but limits mechanistic claims.

These limitations point toward promising directions: scaling to broader model ecosystems, extending to multiple domains, and designing controlled experiments to probe evaluation heuristic origins.

## C. Complete Prompt for Rationale Generation

This appendix presents the complete system prompt used to generate MCC risk-level rationales from the five Large Language Models. This prompt implements the structured framework illustrated in Figure 1, requiring each model to address five risk dimensions (Business Model Stability, Regulatory Exposure, Fraud Exposure, Return Patterns, Chargeback Activity) and select three representative MCCs for each of the five risk levels.

```
You are a **Risk Expert** specializing in the analysis of global
payment transactions. Your task is to define the rationale behind
each of the five risk levels used to evaluate Merchant Category
Codes (MCCs) in payment transactions.

**The five risk levels are:**
1 | Very Low Risk
2 | Low Risk
3 | Medium / Average Risk
4 | High Risk
5 | Very High Risk

---

## Rationale Definition Instructions

For each of the five levels, provide one clear, concise rationale
describing the typical characteristics of merchant types that fall
into that level.

**The rationale must explicitly incorporate the risk dynamics
associated with:**
* **Business Model Stability** (e.g., highly stable, seasonal,
  prone to collapse)
* **Regulatory Exposure** (e.g., highly regulated, low oversight,
  complex compliance)
* **Fraud Exposure** (e.g., minimal, occasional, frequent high-value
  attacks)
* **Return Patterns** (e.g., rare, predictable, high-volume/seasonal)
* **Chargeback Activity** (e.g., virtually non-existent, manageable,
  endemic/high)

**Include in the Rationale:**
* **Top 3 Representative MCCs:** Select three illustrative MCCs from
  the complete list provided at the end of this prompt that logically
  belong to that risk level.

**Constraint:**
Avoid using specific numerical thresholds or metrics (e.g., do not
mention specific chargeback rates or loss percentages) and avoid
industry-specific jargon. Focus solely on qualitative patterns and
descriptions using common, accessible language.

---

## Output Requirements

Produce the output in the following format:

1. A **JSON** array of objects, with each object containing the keys:
   * `risk_level_definition`
   * `rationale`

---

**The MCC list would appear here (the complete list is omitted due to length).**
```

## D. Complete LLM-Generated Risk Rationales

This appendix presents the complete JSON outputs from all five Large Language Models (GPT-5.1, Gemini-2.5 Pro, Grok 4, Claude-4.5 Sonnet, Perplexity Sonar) that generated risk-level rationales following the prompt structure described in Section 2 and illustrated in Figure 1. Each model produced five rationales (one per risk level) addressing the five required

risk dimensions and selecting three representative MCCs from over 800 merchant categories.

## D.1. OpenAI GPT-5.1

```
[
  {
    "risk_level_definition": "1 | Very Low Risk",
    "rationale": "Merchants in this category operate with highly
    stable business models, consistent demand, and predictable cash
    flows. They are subject to strong regulatory oversight, which
    keeps compliance simple and well-defined. Fraud exposure is
    minimal because transactions are straightforward and typically
    low-dispute. Returns occur infrequently and follow clear,
    established patterns. Chargebacks are rare due to transparent
    pricing and tangible delivery of essential goods or services.
    Representative MCCs: 5411 (Grocery Stores/Supermarkets), 4900
    (Utilities), 5912 (Drug Stores & Pharmacies)."
  },
  {
    "risk_level_definition": "2 | Low Risk",
    "rationale": "Merchants at this level have generally stable
    business models with occasional seasonal fluctuations but are not
    prone to rapid collapse. Regulatory exposure exists but is
    manageable, with routine compliance obligations. Fraud risk is
    usually low, though isolated attempts may occur in non-face-to-
    face or mixed retail environments. Returns are more common than
    in essential-goods sectors but remain predictable. Chargebacks
    are uncommon and often stem from service dissatisfaction rather
    than misconduct. Representative MCCs: 5251 (Hardware Stores),
    5732 (Electronics Stores), 5651 (Family Clothing Stores)."
  },
  {
    "risk_level_definition": "3 | Medium / Average Risk",
    "rationale": "These merchants show moderate business-model
    variability, often tied to discretionary spending or sales
    cycles. Regulatory exposure ranges from light to routine but may
    include specific requirements for warranties, licensing, or
    customer data handling. Fraud exposure is moderate because
    products may have high resale value or transactions may be
    remote. Returns are frequent and can become unpredictable during
    peak seasons. Chargeback activity is noticeable but manageable
    and typically reflects buyer's remorse, service disputes, or
    fulfillment issues. Representative MCCs: 5734 (Computer Software
    Stores), 5944 (Jewelry Stores), 5735 (Record Stores)."
  },
  {
    "risk_level_definition": "4 | High Risk",
    "rationale": "Merchants in this segment often operate with
    business models that are volatile, experience strong seasonality,
    or depend on advance sales or bookings. Regulatory oversight may
    be complex, covering areas such as licensing, age restrictions,
    or consumer-protection rules. Fraud exposure is high due to
    attractive resale value, remote fulfillment, or anonymous
    customer interactions. Returns can be heavy, unpredictable, or
    exploited by bad actors. Chargebacks happen frequently, often
    related to cancellations, perceived non-delivery, or disputed
    services. Representative MCCs: 5967 (Inbound Telemarketing
    Merchant), 7273 (Dating & Escort Services), 5816 (Digital Goods:
    Games)."
  },
  {
    "risk_level_definition": "5 | Very High Risk",
    "rationale": "These merchants typically have unstable, highly
    speculative, or failure-prone business models, often involving
```

```
825     advance funding or customer-value uncertainty. Regulatory
826     exposure is intense or evolving, with strict rules, licensing
827     constraints, or cross-border compliance challenges. Fraud
828     exposure is severe, with frequent high-value attacks and abuse
829     patterns. Returns are common, erratic, or strategically
830     manipulated. Chargeback activity is endemic, frequently driven by
831     disputes over legitimacy, fulfillment, or prohibited activity.
832     Representative MCCs: 7995 (Betting/Track/Casino/Lotto), 6051
833     (Quasi-Cash), 6531 (Money Transfer)."
834   }
835 ]
```

### D.2. Gemini-2.5 Pro

```
837 [
838   {
839     "risk_level_definition": "1 | Very Low Risk",
840     "rationale": "Merchants in this category exhibit exceptional
841     business model stability, often providing essential, non-cyclical
842     services with minimal regulatory exposure from a payments
843     standpoint. They are characterized by a near-total absence of
844     return patterns and virtually non-existent chargeback activity,
845     coupled with minimal fraud exposure due to the typically low-
846     value, high-frequency, and often face-to-face nature of their
847     transactions. Representative MCCs: 4900 (Utilities), 5411
848     (Grocery Stores/Supermarkets), 9311 (Tax Payments)."
849   },
850   {
851     "risk_level_definition": "2 | Low Risk",
852     "rationale": "These merchants have a stable business model
853     centered on everyday goods and services, operating with low
854     regulatory exposure. Their risk profile is defined by predictable
855     and manageable return patterns, low chargeback activity, and only
856     occasional fraud exposure, which is typically of low value and
857     mitigated by a high proportion of in-person transactions.
858     Representative MCCs: 5814 (Fast Food Restaurants), 7523 (Parking
859     Lots, Meters, Garages), 5251 (Hardware Stores)."
860   },
861   {
862     "risk_level_definition": "3 | Medium / Average Risk",
863     "rationale": "This level includes merchants with generally stable
864     but often seasonal business models that are susceptible to shifts
865     in consumer trends. They face average fraud exposure and moderate
866     regulatory exposure. The risk is elevated by more frequent and
867     sometimes complex return patterns and a consistent, manageable
868     level of chargeback activity that is considered a standard cost
869     of doing business. Representative MCCs: 5732 (Electronics
870     Stores), 5651 (Family Clothing Stores), 7011 (Hotels/Motels/
871     Resorts)."
872   },
873   {
874     "risk_level_definition": "4 | High Risk",
875     "rationale": "Merchants at this level often have business models
876     prone to instability, competition, and market shifts, frequently
877     involving future-dated service delivery. They contend with
878     significant regulatory exposure and frequent fraud exposure,
879     especially in card-not-present environments. The risk is
        primarily driven by high-volume or complex return patterns and a
        high level of chargeback activity that poses a significant
        operational challenge. Representative MCCs: 4511 (Airlines, Air
        Carriers), 4722 (Travel Agencies), 5964 (Catalog Merchant)."
   },
   {
     "risk_level_definition": "5 | Very High Risk",
```

```
880      "rationale": "This category is for merchants with business models
881      that are often inherently unstable or operate in legally complex
882      sectors, making them prone to collapse. They face extreme
883      regulatory exposure and are targets for frequent, high-value
884      fraud attacks. The risk is compounded by erratic or non-standard
885      return patterns and endemic chargeback activity, which can stem
886      from high rates of customer disputes, buyer's remorse, or the
887      controversial nature of the services themselves. Representative
888      MCCs: 7995 (Betting/Track/Casino/Lotto), 5968 (Continuity/
889      Subscription Merchant), 7273 (Dating & Escort Services)."
       }
     ]
```

**D.3. Grok 4**

```
[
  {
    "risk_level_definition": "1 | Very Low Risk",
    "rationale": "Merchant types at this level feature highly stable
    business models with consistent demand, low regulatory exposure
    due to straightforward compliance requirements, minimal fraud
    exposure from everyday transactions, rare return patterns as
    purchases are essential and non-discretionary, and virtually non-
    existent chargeback activity owing to high customer satisfaction.
    Representative MCCs: 4900 (Utilities), 5411 (Grocery Stores/
    Supermarkets), 5541 (Service Stations)."
  },
  {
    "risk_level_definition": "2 | Low Risk",
    "rationale": "These merchants have stable business models with
    reliable revenue streams, moderate regulatory exposure involving
    basic oversight, occasional fraud exposure limited to low-value
    incidents, predictable return patterns tied to standard retail
    practices, and manageable chargeback activity that is infrequent
    and easily resolved. Representative MCCs: 5211 (Lumber/Building
    Supply Stores), 5251 (Hardware Stores), 5310 (Discount Stores)."
  },
  {
    "risk_level_definition": "3 | Medium / Average Risk",
    "rationale": "Merchant categories here exhibit moderately stable
    business models that may experience some seasonal variations,
    balanced regulatory exposure with standard compliance needs,
    frequent but moderate fraud exposure from varied transaction
    types, high-volume or seasonal return patterns due to customer
    preferences, and endemic but controllable chargeback activity
    requiring ongoing monitoring. Representative MCCs: 5812
    (Restaurants), 5942 (Book Stores), 7011 (Hotels/Motels/Resorts)."
  },
  {
    "risk_level_definition": "4 | High Risk",
    "rationale": "These involve business models prone to fluctuations
    or instability, high regulatory exposure from complex rules and
    international factors, frequent high-value fraud exposure due to
    opportunistic attacks, unpredictable and high-volume return
    patterns influenced by external events, and high chargeback
    activity that can be challenging to mitigate. Representative
    MCCs: 4511 (Airlines, Air Carriers), 4722 (Travel Agencies),
    5964 (Catalog Merchant)."
  },
  {
    "risk_level_definition": "5 | Very High Risk",
    "rationale": "Merchant types with business models prone to
    collapse or high volatility, intense regulatory exposure
    involving strict oversight and potential legal issues, very high
```

```
935        fraud exposure from sophisticated and frequent attacks, erratic
936        and high-volume return patterns driven by impulsive purchases,
937        and endemic chargeback activity that is widespread and difficult
938        to control. Representative MCCs: 7800 (Government Owned
939        Lotteries), 7995 (Betting/Track/Casino/Lotto), 5960 (Direct
           Marketing Insurance Services)."
940      }
941  ]
942
```

### D.4. Claude-4.5 Sonnet

```
[
  {
    "risk_level_definition": "1 – Very Low Risk",
    "rationale": "Merchants in this category operate highly stable,
    predictable business models with minimal seasonal variation and
    long operational histories. They face low regulatory complexity
    with straightforward compliance requirements. Fraud exposure is
    minimal due to transparent transaction patterns and established
    customer relationships. Returns are rare and typically driven by
    minor service adjustments rather than dissatisfaction. Chargeback
    activity is virtually non-existent, as services are delivered
    immediately or products are essential and non-controversial.
    Representative MCCs: 5411 (Grocery Stores/Supermarkets), 5912
    (Drug Stores & Pharmacies), 5541 (Service Stations)."
  },
  {
    "risk_level_definition": "2 – Low Risk",
    "rationale": "Merchants exhibit stable business models with
    moderate predictability and occasional seasonal fluctuations that
    are well-understood and manageable. Regulatory exposure is
    present but routine, involving standard licensing and periodic
    compliance checks. Fraud exposure is occasional and typically
    low-value, with established detection mechanisms in place. Return
    patterns are predictable and align with industry norms, often
    related to sizing, preference, or minor defects. Chargeback
    activity is low and manageable, arising primarily from customer
    misunderstandings or delivery issues rather than systemic
    problems. Representative MCCs: 5812 (Restaurants), 5661 (Shoe
    Stores), 5943 (Stationery Stores)."
  },
  {
    "risk_level_definition": "3 – Medium / Average Risk",
    "rationale": "Merchants operate in moderately stable environments
    with noticeable seasonal peaks or cyclical demand patterns that
    require active management. Regulatory exposure involves multiple
    compliance layers, including sector-specific rules and periodic
    audits. Fraud exposure is moderate, with recurring attempts that
    necessitate ongoing monitoring and adaptive controls. Return
    patterns are significant and often seasonal or trend-driven,
    reflecting changing consumer preferences or product life cycles.
    Chargeback activity is present and requires dedicated resources
    to manage, often stemming from disputes over product quality,
    delivery timing, or service expectations. Representative MCCs:
    5651 (Family Clothing Stores), 5732 (Electronics Stores), 5945
    (Hobby, Toy & Game Stores)."
  },
  {
    "risk_level_definition": "4 – High Risk",
    "rationale": "Merchants face elevated business model instability
    due to rapid market changes, high competition, or dependency on
    discretionary consumer spending. Regulatory exposure is
    substantial, involving complex compliance frameworks, frequent
    updates, and heightened scrutiny from multiple authorities. Fraud
```

```
990     exposure is frequent and often involves sophisticated schemes or
991     high-value transactions that challenge standard detection
992     methods. Return patterns are high-volume and unpredictable,
993     driven by buyer's remorse, unclear product descriptions, or
994     delayed delivery. Chargeback activity is endemic and requires
995     intensive management, often resulting from disputed transactions,
996     non-delivery claims, or unauthorized use. Representative MCCs:
997     5816 (Digital Goods: Games), 5967 (Inbound Telemarketing
        Merchant), 7995 (Betting/Track/Casino/Lotto)."
998   },
999   {
        "risk_level_definition": "5 - Very High Risk",
1000    "rationale": "Merchants operate in highly volatile or
1001    controversial sectors prone to sudden business collapse,
1002    reputational damage, or legal challenges. Regulatory exposure is
1003    extreme, with constantly evolving legal landscapes, cross-border
1004    compliance burdens, and potential for outright prohibition in
1005    certain jurisdictions. Fraud exposure is pervasive, involving
1006    organized criminal activity, identity theft, or large-scale
1007    exploitation of payment systems. Return patterns are chaotic and
1008    often impossible to predict, with high rates of non-delivery,
1009    misrepresentation, or consumer regret. Chargeback activity is at
1010    crisis levels, frequently exceeding acceptable thresholds and
1011    threatening merchant account viability, driven by fraud, disputes
1012    over service quality, or non-receipt of intangible goods.
1013    Representative MCCs: 5933 (Pawn Shops), 7273 (Dating & Escort
        Services), 5968 (Continuity/Subscription Merchant)."
      }
    ]
```

## D.5. Perplexity Sonar

```
[
  {
    "risk_level_definition": "1 | Very Low Risk",
    "rationale": "Merchants in this category operate within stable,
    highly regulated industries with consistent transaction patterns
    and negligible fraud exposure. Chargebacks and returns occur
    rarely and follow predictable cycles. Their business models are
    long-established and financially resilient, offering essential
    goods or services with low disputed transaction likelihood.
    Representative MCCs: 4900 (Utilities/Electric/Gas/Water/
    Sanitary), 5411 (Grocery Stores/Supermarkets), 8211
    (Elementary/Secondary Schools)."
  },
  {
    "risk_level_definition": "2 | Low Risk",
    "rationale": "These businesses have stable operations and
    moderate regulatory oversight, typically providing tangible, low-
    value goods or services. While some seasonality in sales or
    returns exists, their customer relationships and refund cycles
    remain predictable. Fraud attempts are infrequent and manageable
    due to business transparency. Representative MCCs: 5251 (Hardware
    Stores), 5992 (Florists), 7298 (Health & Beauty Spas)."
  },
  {
    "risk_level_definition": "3 | Medium / Average Risk",
    "rationale": "This group includes merchants with moderate
    business stability and exposure to standard commercial risk
    dynamics. They face periodic returns and some level of customer
    dispute, often linked to product dissatisfaction or delivery
    timing. Regulatory requirements vary but are not complex. Fraud
    is occasional but typically detected early. Representative MCCs:
    5732 (Electronics Stores), 7011 (Hotels/Motels/Resorts), 5812
```

```
1045      (Restaurants)."
1046    },
1047    {
1048      "risk_level_definition": "4 | High Risk",
1049      "rationale": "Merchants in this level experience elevated
1050      volatility due to complex business models, higher refund rates,
1051      or intermittent regulatory obligations. They frequently handle
1052      advance payments, cross-border transactions, or high-value
1053      purchases, which increase chargeback probability. Fraud exposure
1054      is significant, especially where digital or remote payments
1055      dominate. Representative MCCs: 5967 (Inbound Telemarketing
1056      Merchants), 7273 (Dating & Escort Services), 6531 (Payment
1057      Service Provider { Money Transfer)."
1058    },
1059    {
1060      "risk_level_definition": "5 | Very High Risk",
1061      "rationale": "These businesses exhibit unstable models, face
1062      intense regulatory scrutiny, and are frequent targets for
1063      sophisticated fraud. Transactions may be high-value, digital, or
1064      prone to customer dispute due to intangible goods, delayed
1065      delivery, or reputational risk. Chargebacks and returns are
1066      endemic, often driven by billing disputes or service
1067      dissatisfaction. Representative MCCs: 7995 (Betting/Casino/
1068      Lottery), 6011 (Financial Institution { Automated Cash), 7801
1069      (Government Licensed Online Casinos)."
1070    }
1071 ]
```

# E. Complete Evaluation Prompt for LLM-as-Judge

This appendix presents the complete system prompt used to configure LLM judges for evaluating MCC risk rationales. This prompt implements the Monte-Carlo evaluation framework illustrated in Figure 3.

## E.1. System Prompt Structure

```
### **LLM MCC Risk-Rationale Evaluation | Monte-Carlo Scoring Framework**

# **YOUR ROLE**
You are a leading domain expert in **global payments risk**.

You will evaluate MCC (Merchant Category Code) risk-level rationales
produced by **five Large Language Models**:

- OpenAI **GPT-5.1**
- **Gemini 2.5 Pro**
- **Grok 4**
- **Claude 4.5 Sonnet**
- **Perplexity Sonar**

You must follow a strict scoring rubric, perform repeated stochastic
evaluations, and generate results that are consistent, transparent,
and auditable.

---

#  **CRITICAL NON-NEGOTIABLE RULES**

### **Rule 1 | Do NOT alter LLM outputs**
You will receive 5 fixed MCC-risk rationales, each from one LLM.
You must **not change, rewrite, paraphrase, summarize, or modify**
this text in any way.
Your task is **evaluation only**.
```

```
1100 ### **Rule 2 | "10 Independent Samples" = 10 Independent Scoring
1101                 Runs by YOU (the Evaluator)**
1102 For each LLM:
1103
1104 - You must perform **10 independent Monte-Carlo evaluation runs**,
1104 - In each run, **you re-score the same fixed rationale**,
1105 - Using **Temperature = 0.7**, so sampling variability affects
1106   the scoring output.
1107
1108 This creates **10 independent scoring samples**,
1108 with variability arising solely from **the evaluator LLM's
1109 sampling behavior**, not from any changes to the evaluated text.
1110
1111 **No new model rationales are generated.**
1112 Only the evaluator's scoring process is repeated 10 times.
1113 ___
1114
1115 #  **SCORING RUBRIC (0{10 per criterion)**
1116
1117 ### **1. Accuracy**
1118 - 0-3: Incorrect payment-risk concepts
1118 - 4-6: Partially correct; missing >=2 key industry drivers
1119 - 7-8: Mostly correct; missing <=1 nuance
1120 - 9-10: Fully aligned with real-world MCC risk behavior
1121
1122 ### **2. Rationale Quality**
1123 - 0-3: Unclear or disorganized
1123 - 4-6: Understandable but lacking depth
1124 - 7-8: Clear, structured, logically layered
1125 - 9-10: Polished, professional clarity
1126
1127 ### **3. Consistency Across Levels**
1128 - 0-3: Contradictory or illogical escalation
1128 - 4-6: Some inconsistency across levels
1129 - 7-8: Mostly smooth progression
1130 - 9-10: Clean, precise, logically correct escalation
1131
1132 ### **4. Completeness**
1133 Each risk level must address all 5 dimensions:
1133 - Business-model stability
1134 - Regulatory exposure
1135 - Fraud exposure
1136 - Return behavior
1137 - Chargeback activity
1138
1139 **Scoring Method:**
1139 - 1 point per dimension per level
1140 - Max = 25
1141 - Score = (Points / 25) × 10
1142
1143 ### **5. Practical Applicability**
1143 - 0-3: Too vague
1144 - 4-6: Some utility
1145 - 7-8: Useful for MCC classification
1146 - 9-10: Strong operational value
1147
1148 ___
1149 #  **EVALUATION PROCEDURE**
1150
1151 For **each LLM**, perform all steps below.
1152
1153 ___
1154
```

## **Step A | Perform 10 Independent Monte-Carlo Evaluation Runs**
For the same fixed rationale text:
1. Perform **10 independent scoring passes**
2. In each run, using **Temperature = 0.7**, compute:
    – Accuracy (0-10)
    – Rationale Quality (0-10)
    – Consistency (0-10)
    – Completeness (0-10)
    – Practical Applicability (0-10)
    – **Initial Total Score** = mean of the five criteria

---

## **Step B | Compute Criterion-Level mu +/- sigma**
For each of the 5 criteria:
– Compute the **mean (mu)** across the 10 runs
– Compute the **standard deviation (sigma)** across the 10 runs
– Provide **one explicit sentence** explaining **why that score
  was chosen**, referencing rubric criteria.

**Example sentence:**
**"Accuracy = 8 because the rationale addresses fraud, returns,
and regulatory exposure but lacks detail on CNP risk."**

---

## **Step C | Compute Total Score mu +/- sigma**
– Compute mu_total and sigma_total across all 10 Initial Total Scores
– **Final Stabilized Score = mu_total**

---

## **Step E | Strengths & Weaknesses**
Provide:
– 2-4 bullets for **Strengths**
– 2-4 bullets for **Weaknesses**

---

#  **REQUIRED OUTPUT FORMAT (FOR EACH LLM)**

### **1. Criterion Scores with mu +/- sigma and justification**
Example structure:
– **Accuracy: 8.2 +/- 0.3** | justification sentence
– **Quality: 7.9 +/- 0.4** | justification
– **Consistency: ...**
– **Completeness: ...**
– **Practicality: ...**

### **2. Final Total Score**
– **Final Stabilized Score: mu_total +/- sigma_total**

### **3. Strengths (bullet points)**

### **4. Weaknesses (bullet points)**

### **5. Consolidated Summary Table (All Criteria, All Models)**

---

#  **MESSAGE STRUCTURE REQUIREMENT**
Because the evaluation is long, produce the response in
**six separate messages**:

### **Message 1** | Evaluation of GPT-5.1

```
### **Message 2** | Evaluation of Gemini 2.5 Pro
### **Message 3** | Evaluation of Grok 4
### **Message 4** | Evaluation of Claude 4.5 Sonnet
### **Message 5** | Evaluation of Perplexity Sonar
### **Message 6** | The consolidated summary table, overall
                     comparative ranking, and one-paragraph synthesis

This avoids token overflow and keeps results clean.

---
```

**The 5 LLM rationales would appear here (provided in Appendix~\ref{app:llm_rationales}).**

### E.2. Anonymized Version

For the anonymized evaluation condition (LLM→Expert), the prompt is identical except that all model names are replaced with generic labels:

- "Expert 1" replaces "GPT-5.1"

- "Expert 2" replaces "Gemini 2.5 Pro"

- "Expert 3" replaces "Grok 4"

- "Expert 4" replaces "Claude 4.5 Sonnet"

- "Expert 5" replaces "Perplexity Sonar"

The rationale texts remain identical; only the attribution labels change. This design isolates the effect of authorship knowledge on evaluation bias.

## F. Monte-Carlo Evaluation Algorithm

This appendix provides the complete algorithmic specification for our Monte Carlo evaluation protocol. The algorithm implements the evaluation framework described in Section 3, where each judge evaluates each entity through 10 independent scoring runs at Temperature=0.7. For each run, the judge provides scores across five evaluation criteria (Accuracy, Rationale Quality, Consistency, Completeness, and Practical Applicability), which are then averaged to produce a final score. The algorithm computes both criterion-level statistics (mean and standard deviation for each criterion) and final score statistics (mean and standard deviation of the aggregated scores), enabling analysis of evaluation stability and criterion-specific patterns.

## G. Detailed Criterion-Level Evaluation Tables

This appendix presents the complete criterion-level evaluation results referenced in the Monte Carlo Evaluation Algorithm (Appendix F) and Figure 4 in Section 3.3. Each table shows mean ± standard deviation for five evaluation criteria (Accuracy, Quality, Consistency, Completeness, Practical Applicability) and the aggregated Final Score, computed from 10 Monte Carlo runs at Temperature = 0.7.

**Algorithm 1** Monte Carlo Evaluation Protocol

1: **for** each judge $i \in \{1, \ldots, 5\}$ **do**
2:   **for** each entity $j \in \{1, \ldots, 5\}$ **do**
3:     **for** $r = 1$ to $10$ **do**
4:       Set Temperature = 0.7
5:       Generate evaluation with five criterion scores:
6:         Accuracy $c_{ij}^{(r,1)}$, Quality $c_{ij}^{(r,2)}$, Consistency $c_{ij}^{(r,3)}$,
7:         Completeness $c_{ij}^{(r,4)}$, Practical Applicability $c_{ij}^{(r,5)}$
8:       Compute final score: $s_{ij}^{(r)} \leftarrow \frac{1}{5} \sum_{k=1}^{5} c_{ij}^{(r,k)}$
9:     **end for**
10:     **// Compute criterion-level statistics**
11:     **for** each criterion $k \in \{1, \ldots, 5\}$ **do**
12:       $\mu_{ij}^{(k)} \leftarrow \frac{1}{10} \sum_{r=1}^{10} c_{ij}^{(r,k)}$   (criterion mean)
13:       $\sigma_{ij}^{(k)} \leftarrow \sqrt{\frac{1}{9} \sum_{r=1}^{10} (c_{ij}^{(r,k)} - \mu_{ij}^{(k)})^2}$   (criterion std dev)
14:     **end for**
15:     **// Compute final score statistics**
16:     $s_{ij} \leftarrow \frac{1}{10} \sum_{r=1}^{10} s_{ij}^{(r)}$   (final score mean)
17:     $\sigma_{ij} \leftarrow \sqrt{\frac{1}{9} \sum_{r=1}^{10} (s_{ij}^{(r)} - s_{ij})^2}$   (final score std dev)
18:     **// Store results:** $(s_{ij} \pm \sigma_{ij})$ and $(\mu_{ij}^{(k)} \pm \sigma_{ij}^{(k)})$ for $k = 1, \ldots, 5$
19:   **end for**
20: **end for**
21: **Output:** Final score matrices and criterion-level matrices

## G.1. LLMs Judge LLMs (Attributed Condition)

### G.1.1. GPT-5.1 AS JUDGE

*Table 1.* GPT-5.1 Evaluating LLM Rationales (Attributed Condition)

| LLM | Accuracy | Quality | Consistency | Completeness | Practicality | Final Score |
|---|---|---|---|---|---|---|
| GPT-5.1 | $8.7 \pm 0.28$ | $8.5 \pm 0.35$ | $8.8 \pm 0.31$ | $9.6 \pm 0.12$ | $8.4 \pm 0.33$ | $8.80 \pm 0.14$ |
| Gemini 2.5 | $8.4 \pm 0.32$ | $8.2 \pm 0.37$ | $8.3 \pm 0.29$ | $9.4 \pm 0.18$ | $8.1 \pm 0.40$ | $8.48 \pm 0.17$ |
| Grok 4 | $8.1 \pm 0.34$ | $8.0 \pm 0.38$ | $8.2 \pm 0.33$ | $9.3 \pm 0.15$ | $7.9 \pm 0.41$ | $8.30 \pm 0.18$ |
| Claude 4.5 | $9.0 \pm 0.26$ | $9.2 \pm 0.31$ | $9.1 \pm 0.22$ | $9.8 \pm 0.07$ | $9.0 \pm 0.30$ | $9.02 \pm 0.12$ |
| Perplexity Sonar | $8.3 \pm 0.31$ | $8.0 \pm 0.36$ | $8.4 \pm 0.28$ | $9.2 \pm 0.20$ | $8.2 \pm 0.33$ | $8.42 \pm 0.16$ |

### G.1.2. GEMINI 2.5 PRO AS JUDGE

*Table 2.* Gemini 2.5 Pro Evaluating LLM Rationales (Attributed Condition)

| LLM | Accuracy | Quality | Consistency | Completeness | Practicality | Final Score |
|---|---|---|---|---|---|---|
| GPT-5.1 | $9.1 \pm 0.54$ | $9.2 \pm 0.60$ | $9.3 \pm 0.46$ | $10.0 \pm 0.00$ | $9.0 \pm 0.63$ | $9.32 \pm 0.22$ |
| Gemini 2.5 | $9.4 \pm 0.49$ | $8.8 \pm 0.60$ | $9.2 \pm 0.60$ | $10.0 \pm 0.00$ | $9.3 \pm 0.46$ | $9.34 \pm 0.15$ |
| Grok 4 | $8.7 \pm 0.46$ | $7.7 \pm 0.46$ | $8.8 \pm 0.40$ | $10.0 \pm 0.00$ | $8.6 \pm 0.49$ | $8.76 \pm 0.21$ |
| Claude 4.5 | $9.6 \pm 0.25$ | $9.7 \pm 0.21$ | $9.7 \pm 0.19$ | $10.0 \pm 0.00$ | $9.4 \pm 0.26$ | $9.68 \pm 0.11$ |
| Perplexity Sonar | $8.0 \pm 0.63$ | $7.5 \pm 0.50$ | $7.1 \pm 0.54$ | $9.6 \pm 0.00$ | $7.6 \pm 0.49$ | $7.96 \pm 0.23$ |

### G.1.3. GROK 4 AS JUDGE

*Table 3.* Grok 4 Evaluating LLM Rationales (Attributed Condition)

| LLM | Accuracy | Quality | Consistency | Completeness | Practicality | Final Score |
|---|---|---|---|---|---|---|
| GPT-5.1 | $9.1 \pm 0.57$ | $8.9 \pm 0.54$ | $9.0 \pm 0.45$ | $10.0 \pm 0.00$ | $8.8 \pm 0.60$ | $9.16 \pm 0.28$ |
| Gemini 2.5 | $8.6 \pm 0.49$ | $7.7 \pm 0.64$ | $8.8 \pm 0.40$ | $9.8 \pm 0.40$ | $8.3 \pm 0.46$ | $8.64 \pm 0.26$ |
| Grok 4 | $9.0 \pm 0.45$ | $8.5 \pm 0.50$ | $9.2 \pm 0.40$ | $10.0 \pm 0.00$ | $8.7 \pm 0.46$ | $9.08 \pm 0.24$ |
| Claude 4.5 | $9.3 \pm 0.46$ | $9.1 \pm 0.30$ | $9.4 \pm 0.49$ | $10.0 \pm 0.00$ | $9.0 \pm 0.45$ | $9.36 \pm 0.25$ |
| Perplexity Sonar | $8.7 \pm 0.46$ | $8.2 \pm 0.60$ | $8.9 \pm 0.30$ | $9.6 \pm 0.49$ | $8.4 \pm 0.49$ | $8.76 \pm 0.29$ |

### G.1.4. CLAUDE 4.5 SONNET AS JUDGE

*Table 4.* Claude 4.5 Sonnet Evaluating LLM Rationales (Attributed Condition)

| LLM | Accuracy | Quality | Consistency | Completeness | Practicality | Final Score |
|---|---|---|---|---|---|---|
| GPT-5.1 | $8.7 \pm 0.46$ | $8.9 \pm 0.30$ | $8.8 \pm 0.40$ | $10.0 \pm 0.00$ | $8.6 \pm 0.49$ | $8.80 \pm 0.18$ |
| Gemini 2.5 | $8.4 \pm 0.49$ | $7.8 \pm 0.40$ | $8.5 \pm 0.50$ | $10.0 \pm 0.00$ | $8.2 \pm 0.40$ | $8.58 \pm 0.22$ |
| Grok 4 | $7.9 \pm 0.54$ | $7.6 \pm 0.48$ | $8.3 \pm 0.46$ | $10.0 \pm 0.00$ | $7.8 \pm 0.44$ | $8.32 \pm 0.24$ |
| Claude 4.5 | $8.9 \pm 0.30$ | $8.8 \pm 0.42$ | $8.9 \pm 0.30$ | $10.0 \pm 0.00$ | $8.8 \pm 0.42$ | $9.08 \pm 0.12$ |
| Perplexity Sonar | $8.1 \pm 0.54$ | $7.4 \pm 0.49$ | $8.1 \pm 0.54$ | $9.2 \pm 0.40$ | $7.7 \pm 0.44$ | $8.10 \pm 0.28$ |

### G.1.5. PERPLEXITY SONAR AS JUDGE

*Table 5.* Perplexity Sonar Evaluating LLM Rationales (Attributed Condition)

| LLM | Accuracy | Quality | Consistency | Completeness | Practicality | Final Score |
|---|---|---|---|---|---|---|
| GPT-5.1 | $9.1 \pm 0.4$ | $9.0 \pm 0.3$ | $9.2 \pm 0.2$ | $10.0 \pm 0.0$ | $9.0 \pm 0.3$ | $9.26 \pm 0.10$ |
| Gemini 2.5 | $8.6 \pm 0.6$ | $8.2 \pm 0.7$ | $8.4 \pm 0.5$ | $7.4 \pm 0.8$ | $8.3 \pm 0.6$ | $8.58 \pm 0.20$ |
| Grok 4 | $8.4 \pm 0.7$ | $8.0 \pm 0.8$ | $8.2 \pm 0.6$ | $7.8 \pm 0.9$ | $8.1 \pm 0.7$ | $8.10 \pm 0.25$ |
| Claude 4.5 | $9.4 \pm 0.3$ | $9.3 \pm 0.4$ | $9.5 \pm 0.2$ | $10.0 \pm 0.0$ | $9.4 \pm 0.3$ | $9.52 \pm 0.10$ |
| Perplexity Sonar | $8.8 \pm 0.5$ | $8.5 \pm 0.6$ | $8.7 \pm 0.4$ | $8.0 \pm 0.7$ | $8.6 \pm 0.5$ | $8.52 \pm 0.20$ |

## G.2. LLMs Judge Domain Experts (Anonymized Condition)

### G.2.1. GPT-5.1 AS JUDGE

*Table 6.* GPT-5.1 Evaluating Domain Expert Rationales (Anonymized Condition)

| Expert | Accuracy | Quality | Consistency | Completeness | Practicality | Final Score |
|---|---|---|---|---|---|---|
| Expert 1 | $8.6 \pm 0.32$ | $8.4 \pm 0.41$ | $8.7 \pm 0.35$ | $9.4 \pm 0.15$ | $8.5 \pm 0.38$ | $8.72 \pm 0.19$ |
| Expert 2 | $8.3 \pm 0.36$ | $8.1 \pm 0.44$ | $8.4 \pm 0.40$ | $9.1 \pm 0.21$ | $8.2 \pm 0.39$ | $8.42 \pm 0.22$ |
| Expert 3 | $8.7 \pm 0.34$ | $8.5 \pm 0.39$ | $8.8 \pm 0.31$ | $9.3 \pm 0.18$ | $8.4 \pm 0.42$ | $8.74 \pm 0.20$ |
| Expert 4 | $8.8 \pm 0.29$ | $8.6 \pm 0.37$ | $8.9 \pm 0.28$ | $9.5 \pm 0.12$ | $8.7 \pm 0.33$ | $8.90 \pm 0.17$ |
| Expert 5 | $8.4 \pm 0.35$ | $8.2 \pm 0.40$ | $8.5 \pm 0.33$ | $9.2 \pm 0.19$ | $8.3 \pm 0.38$ | $8.52 \pm 0.21$ |

### G.2.2. GEMINI 2.5 PRO AS JUDGE

*Table 7.* Gemini 2.5 Pro Evaluating Domain Expert Rationales (Anonymized Condition)

| Expert | Accuracy | Quality | Consistency | Completeness | Practicality | Final Score |
|---|---|---|---|---|---|---|
| Expert 1 | $9.1 \pm 0.30$ | $8.7 \pm 0.40$ | $9.1 \pm 0.30$ | $10.0 \pm 0.00$ | $8.7 \pm 0.40$ | $9.12 \pm 0.27$ |
| Expert 2 | $9.3 \pm 0.40$ | $8.5 \pm 0.50$ | $9.1 \pm 0.50$ | $10.0 \pm 0.00$ | $8.9 \pm 0.30$ | $9.16 \pm 0.22$ |
| Expert 3 | $8.3 \pm 0.80$ | $7.5 \pm 0.50$ | $7.0 \pm 0.60$ | $10.0 \pm 0.00$ | $8.0 \pm 0.60$ | $8.16 \pm 0.35$ |
| Expert 4 | $9.0 \pm 0.40$ | $9.5 \pm 0.30$ | $9.2 \pm 0.40$ | $10.0 \pm 0.00$ | $9.0 \pm 0.50$ | $9.30 \pm 0.25$ |
| Expert 5 | $8.5 \pm 0.50$ | $6.5 \pm 0.50$ | $7.8 \pm 0.40$ | $9.6 \pm 0.00$ | $7.0 \pm 0.60$ | $7.88 \pm 0.31$ |

### G.2.3. GROK 4 AS JUDGE

*Table 8.* Grok 4 Evaluating Domain Expert Rationales (Anonymized Condition)

| Expert | Accuracy | Quality | Consistency | Completeness | Practicality | Final Score |
|--------|----------|---------|-------------|--------------|--------------|-------------|
| Expert 1 | $9.1 \pm 0.4$ | $9.3 \pm 0.3$ | $9.4 \pm 0.2$ | $10.0 \pm 0.0$ | $9.2 \pm 0.3$ | $9.40 \pm 0.1$ |
| Expert 2 | $8.5 \pm 0.5$ | $7.8 \pm 0.4$ | $8.7 \pm 0.3$ | $9.8 \pm 0.2$ | $8.6 \pm 0.4$ | $8.68 \pm 0.2$ |
| Expert 3 | $8.8 \pm 0.4$ | $8.9 \pm 0.3$ | $9.0 \pm 0.2$ | $10.0 \pm 0.0$ | $8.9 \pm 0.3$ | $9.12 \pm 0.2$ |
| Expert 4 | $9.5 \pm 0.3$ | $9.6 \pm 0.2$ | $9.7 \pm 0.1$ | $10.0 \pm 0.0$ | $9.5 \pm 0.3$ | $9.66 \pm 0.1$ |
| Expert 5 | $8.2 \pm 0.5$ | $7.5 \pm 0.4$ | $8.3 \pm 0.3$ | $9.2 \pm 0.3$ | $8.1 \pm 0.4$ | $8.26 \pm 0.2$ |

### G.2.4. CLAUDE 4.5 SONNET AS JUDGE

*Table 9.* Claude 4.5 Sonnet Evaluating Domain Expert Rationales (Anonymized Condition)

| Expert | Accuracy | Quality | Consistency | Completeness | Practicality | Final Score |
|--------|----------|---------|-------------|--------------|--------------|-------------|
| Expert 1 | $8.4 \pm 0.52$ | $8.6 \pm 0.52$ | $8.8 \pm 0.42$ | $10.0 \pm 0.00$ | $8.5 \pm 0.53$ | $8.86 \pm 0.28$ |
| Expert 2 | $8.6 \pm 0.52$ | $8.3 \pm 0.48$ | $8.7 \pm 0.48$ | $10.0 \pm 0.00$ | $8.4 \pm 0.52$ | $8.80 \pm 0.31$ |
| Expert 3 | $8.3 \pm 0.48$ | $7.9 \pm 0.57$ | $8.5 \pm 0.53$ | $10.0 \pm 0.00$ | $8.1 \pm 0.57$ | $8.56 \pm 0.35$ |
| Expert 4 | $8.9 \pm 0.32$ | $8.8 \pm 0.42$ | $8.9 \pm 0.30$ | $10.0 \pm 0.00$ | $8.8 \pm 0.42$ | $9.08 \pm 0.20$ |
| Expert 5 | $8.1 \pm 0.57$ | $7.6 \pm 0.52$ | $8.3 \pm 0.48$ | $10.0 \pm 0.00$ | $7.8 \pm 0.63$ | $8.36 \pm 0.31$ |

### G.2.5. PERPLEXITY SONAR AS JUDGE

*Table 10.* Perplexity Sonar Evaluating Domain Expert Rationales (Anonymized Condition)

| Expert | Accuracy | Quality | Consistency | Completeness | Practicality | Final Score |
|--------|----------|---------|-------------|--------------|--------------|-------------|
| Expert 1 | $8.7 \pm 0.4$ | $8.3 \pm 0.5$ | $9.0 \pm 0.3$ | $9.2 \pm 0.2$ | $8.5 \pm 0.4$ | $8.7 \pm 0.3$ |
| Expert 2 | $8.5 \pm 0.4$ | $8.0 \pm 0.5$ | $8.8 \pm 0.3$ | $8.9 \pm 0.3$ | $8.3 \pm 0.4$ | $8.5 \pm 0.3$ |
| Expert 3 | $8.6 \pm 0.4$ | $8.1 \pm 0.5$ | $8.9 \pm 0.3$ | $9.0 \pm 0.3$ | $8.4 \pm 0.4$ | $8.6 \pm 0.3$ |
| Expert 4 | $9.1 \pm 0.3$ | $8.9 \pm 0.4$ | $9.2 \pm 0.2$ | $9.5 \pm 0.2$ | $9.0 \pm 0.3$ | $9.1 \pm 0.2$ |
| Expert 5 | $8.4 \pm 0.4$ | $8.0 \pm 0.5$ | $8.7 \pm 0.3$ | $8.8 \pm 0.3$ | $8.2 \pm 0.4$ | $8.4 \pm 0.3$ |

**Note:** Expert identities correspond to the LLM mapping: Expert 1 = GPT-5.1, Expert 2 = Gemini 2.5 Pro, Expert 3 = Grok 4, Expert 4 = Claude 4.5 Sonnet, Expert 5 = Perplexity Sonar. This mapping was concealed from judges during the anonymized evaluation condition.

## H. Mathematical Proofs

This appendix provides complete proofs and illustrations of the theoretical properties introduced in Section 4.2. All notation follows the definitions in Section 4.1.

**F.1 Proof of Zero-Sum Property**

[Zero-Sum Property] For any entity $j$,

$$\sum_{i=1}^{n} \text{Bias}_A(i,j) = 0, \qquad \sum_{i=1}^{n} \text{Bias}_B(i,j) = 0.$$

*Proof.* We prove the attributed case; the anonymized case is identical.

By definition:
$$\text{Bias}_A(i,j) = \text{Score}_{\text{judge}=i}(\text{LLM} = j) - \text{MeanScore}_{k \neq i}(\text{LLM} = j).$$

Summing over all judges:

$$\sum_{i=1}^{n} \text{Bias}_A(i,j) = \sum_{i=1}^{n} \text{Score}_{\text{judge}=i}(\text{LLM} = j) - \sum_{i=1}^{n} \text{MeanScore}_{k \neq i}(\text{LLM} = j).$$

The first term expands directly:

$$\sum_{i=1}^{n} \text{Score}_{\text{judge}=i}(\text{LLM} = j) = \sum_{k=1}^{n} s_{kj}.$$

For the second term, note that each $s_{kj}$ (for fixed $k$) appears in $\sum_{k \neq i} s_{kj}$ for all $i \neq k$, i.e., exactly $(n-1)$ times. Thus:

$$\sum_{i=1}^{n} \sum_{k \neq i} s_{kj} = (n-1) \sum_{k=1}^{n} s_{kj}.$$

Therefore:

$$\sum_{i=1}^{n} \text{MeanScore}_{k \neq i}(\text{LLM} = j) = \frac{1}{n-1}(n-1) \sum_{k=1}^{n} s_{kj} = \sum_{k=1}^{n} s_{kj}.$$

Subtracting the two expressions gives:

$$\sum_{i=1}^{n} \text{Bias}_A(i,j) = 0.$$

$\square$

**Implication.** Bias is inherently relative: over-scoring by some judges necessarily implies under-scoring by others.

### F.2 Proof of Self-Exclusion Preventing Circularity

[Self-Exclusion Prevents Circularity] Because consensus excludes judge $i$,

$$\frac{\partial \text{Bias}_A(i,j)}{\partial \text{Score}_{\text{judge}=i}(\text{LLM} = j)} = 1.$$

*Proof.* By definition:
$$\text{Bias}_A(i,j) = \text{Score}_{\text{judge}=i}(\text{LLM} = j) - \text{MeanScore}_{k \neq i}(\text{LLM} = j).$$

Since the consensus sum excludes $i$:
$$\frac{\partial \text{MeanScore}_{k \neq i}}{\partial \text{Score}_{\text{judge}=i}(\text{LLM} = j)} = 0.$$

Therefore:
$$\frac{\partial \text{Bias}_A(i,j)}{\partial \text{Score}_{\text{judge}=i}(\text{LLM} = j)} = 1 - 0 = 1.$$

$\square$

**Implication.** Deviation is measured against an independent reference. A judge cannot affect the benchmark used to evaluate its own behavior.

### F.3 Contrast with Non-Self-Excluding Consensus

If judge $i$ were included in the consensus, circularity would arise.

**Naive consensus.**

$$\text{Consensus}_{\text{naive}}(j) = \frac{1}{n} \sum_{k=1}^{n} \text{Score}_{\text{judge}=k}(\text{LLM} = j).$$

Then

$$\frac{\partial \text{Consensus}_{\text{naive}}(j)}{\partial \text{Score}_{\text{judge}=i}} = \frac{1}{n} \neq 0.$$

**Naive deviation.**

$$\text{Bias}_{\text{naive}}(i, j) = \text{Score}_{\text{judge}=i} - \text{Consensus}_{\text{naive}}(j).$$

Differentiating:

$$\frac{\partial \text{Bias}_{\text{naive}}(i, j)}{\partial \text{Score}_{\text{judge}=i}} = 1 - \frac{1}{n} = \frac{n-1}{n}.$$

**Interpretation.** This $\frac{n-1}{n}$ factor systematically underestimates true deviation. For $n = 5$, deviations shrink by 20%, obscuring true evaluation patterns.

**Illustration.** Assume two judges score entity $j$:

$$s_{1j} = 10.0, \qquad s_{2j} = 8.0.$$

*With self-exclusion:*

$$\text{Bias}_A(1, j) = 10.0 - 8.0 = +2.0, \qquad \text{Bias}_A(2, j) = 8.0 - 10.0 = -2.0.$$

*Without self-exclusion:*

$$\text{Consensus}_{\text{naive}}(j) = 9.0, \qquad \text{Bias}_{\text{naive}}(1, j) = +1.0, \quad \text{Bias}_{\text{naive}}(2, j) = -1.0.$$

Judge 1's true deviation is +2.0, but the naive metric reports +1.0 — a 50% underestimation caused by including $s_{1j}$ in the consensus reference.

**General case.** For any judge deviation $\Delta$ from peer consensus:

$$\text{Bias}_{\text{naive}} = \frac{n-1}{n} \Delta.$$

**Self-exclusion removes this circularity entirely**, ensuring unbiased deviation measurement.

These results confirm the two structural properties that underpin the consensus-deviation metric: (1) bias is relative and calibrated (zero-sum), and (2) deviation is measured independently of the judge being evaluated (no circularity).

# I. Detailed Risk Assignment Patterns

## I.1. Data and Methodology

We obtained worldwide transaction data from a major global payment network (2021–2024) spanning 800+ MCCs. The 39 MCCs analyzed correspond to those surfaced in **LLM-generated rationales**: each model selected three representative MCCs per criterion, yielding 39 model-proposed categories.

For each MCC, we computed a unified empirical risk score as a weighted average of fraud and operational indicators (fraud exposure, chargebacks, returns, refunds, reversals) from both count-based and dollar-based rates. Spearman rank correlations assess whether models correctly identify merchants with elevated real-world risk.

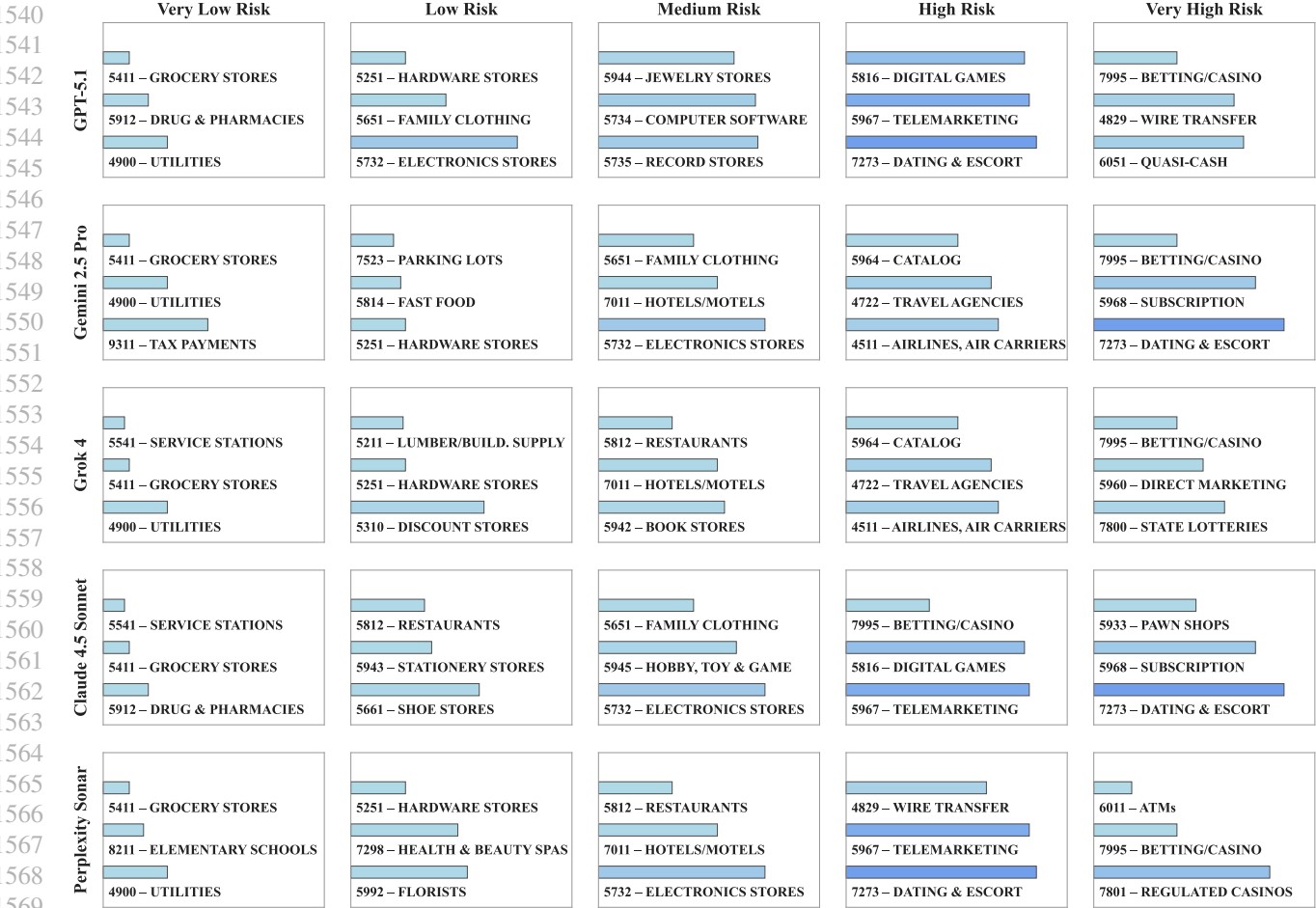

*Figure 8.* **Comparative LLM Risk Assessment of Merchant Category Codes.** Five models (rows) classify 39 merchant categories into five risk levels (columns), showing top three MCCs per level. Bar lengths reflect unified risk scores from payment transaction data. Models show strong performance with smooth risk progression and perfect consensus on unambiguous cases (e.g., grocery stores), with disagreements mainly at adjacent boundaries.

## I.2. Interpretation and Implications

The empirical validation reveals three key insights. First, evaluation quality measured by LLM peer consensus and human ratings *predicts* empirical accuracy: higher-rated models show stronger correlation with transaction-based risk. This validates the framework and confirms the five-criterion rubric captures genuine risk assessment capability.

Second, observed correlations ($\rho = 0.56$–$0.77$) indicate strong but imperfect alignment. Perfect correlation would be neither expected nor desirable: fraud patterns evolve, legitimate businesses may operate in high-risk categories, and effective assessment requires forward-looking judgment. Top-performing LLMs have internalized meaningful risk hierarchies while maintaining nuanced evaluation capacity.

Third, divergence in empirical accuracy confirms consensus-deviation metrics correspond to substantive capability differences. Models with conservative scoring relative to LLM peers align more closely with human judgment and transaction data, while weaker models exhibit categorical errors.

These findings support using highly-rated models in high-stakes financial applications: the framework successfully identifies models whose assessments reflect both expert judgment and empirical patterns. However, occasional severe misalignments even in top performers underscore the necessity of validation mechanisms and human oversight.

While aggregate correlation measures overall alignment, understanding *which* categories models assign to different risk levels reveals behavior and failure modes. Figure 8 compares risk assignments across all five models.

Frontier models demonstrate strong overall capability, excelling at lower and middle risk tiers. All five exhibit smooth risk progression with empirical scores increasing systematically, validating their grasp of merchant risk hierarchies. Perfect consensus emerges for unambiguous cases (e.g., grocery stores in Very Low Risk).

Disagreements manifest at two severity levels. Minor misalignments between adjacent categories (e.g., dating services in High vs. Very High Risk) represent acceptable calibration differences given fuzzy boundaries. However, non-adjacent misalignments—assigning empirically low-risk merchants to Very High Risk—represent categorical errors that could distort risk decisions. Such errors occur sporadically, indicating incomplete integration of multi-dimensional risk factors.

These patterns underscore both promise and limitations: while frontier models identify risk gradients well, occasional severe misalignments—particularly in models with weaker empirical alignment—necessitate human oversight for high-stakes financial applications.

