# OpenReview forum: "Understanding LLM Evaluator Behavior: A Structured Multi‑Evaluator Framework for Merchant Risk Assessment"
_ICML.cc/2026/Conference — Submitted to ICML 2026_

### Official Review · Reviewer_dxup · 2026-03-08

**Soundness:** 1
**Presentation:** 2
**Significance:** 2
**Originality:** 1
**Overall Recommendation:** 2
**Confidence:** 5

**Summary:**

This paper presents a leave-one-out metric for evaluating LLM self-evaluation bias, applied to risk rationales across merchant categories. Five frontier LLMs generate and cross-evaluate outputs under attributed and anonymized conditions, with validation against 26 payment-industry experts and transaction data. The domain application is interesting and the triangulated validation design is careful. However, the technical contribution is thin and the paper's own theoretical results create an internal contradiction with its practical recommendations.

**Compliance With Llm Reviewing Policy:**

Affirmed.

**Final Justification:**

Same as before and I didn't particularly felt the rebuttal engaged with my substantive comments.

**Key Questions For Authors:**

1. *On the internal contradiction*: How do the authors reconcile the ensemble debiasing recommendation with Proposition 4.1, which guarantees zero average bias across judges by construction?
2. *On pool stability*: How sensitive are the reported bias estimates to pool composition? What happens to each model's bias score when one of the five models is removed or replaced? This seems like a critical robustness check given n=5.
3. *On downstream validity*: The core application of LLM scores is to augment or automate human decisions. Is there evidence that bias-corrected scores are better predictors of human judgment than raw scores? The paper shows directional agreement but not that the correction improves downstream utility.
4. *On anonymization*: How much of the attributed vs. anonymized difference survives when stylistic markers beyond the name label are controlled for? Has any attempt been made to test whether the LLMs can identify authorship from the content alone? What if we swap a "fake authorship" name, say, ChatGPT's name onto a report of Claude? Would the LLM-as-a-judge be fooled or stay stable?

**Limitations:**

yes

**Strengths And Weaknesses:**

## Strength
1. The dual attributed/anonymized design cleanly isolates identity-driven versus intrinsic scoring tendencies.
2. Negative self-evaluation bias in high-performing models is a novel finding for MCP applications; most prior work looked at self-preference.

## Weakness
1. The core metric is **not novel**. Computing additive bias relative to the leave-one-out mean is standard in the inter-rater reliability literature (Shrout & Fleiss, 1979) and is equivalent to a simple additive linear model, one of the oldest and most basic model in the peer grading and reviewer calibration literature. The paper presents this without engagement with that prior art, which would substantially contextualize the contribution.
2. The propositions are more elementary algebraic identities than novel theorems. Proposition 4.1 (zero-sum) follows trivially from the definition of deviation from a mean. Proposition 4.2 (derivative=1) is a one-line consequence of self-exclusion. Presenting these as "theoretical guarantees distinguishing it from naive baselines" significantly overstates their depth.
3. **Internal contradiction between Proposition 4.1 and the practical recommendations.** The implications section recommends multi-judge ensembles as a debiasing strategy. But Proposition 4.1 guarantees that the average bias across all judges is exactly zero by construction. Averaging the panel produces zero net bias not because the ensemble is well-calibrated, but because the biases arithmetically cancel within the pool. This directly contradicts the recommendation and is not acknowledged.
4. The metric is **entirely peer-group relative with no external (e.g., human) anchor.** All bias scores are defined relative to the specific 5-model pool chosen. Swap in or out a single model and every number changes. The paper frames the zero-sum property as a desirable feature, but it is actually the clearest symptom of this limitation — the scores are guaranteed to carry no absolute meaning by construction. The paper does not make this limitation sufficiently clear, and some claims about bias magnitude implicitly treat the scores as if they were absolute.
5. The human expert data is underutilized. With 26 expert scores in hand, the authors have a genuine gold standard that could anchor the LLM scores absolutely rather than relatively. prediction-powered-inference-style calibration against the expert scores would transform the metric from peer-relative to human-anchored, giving the bias estimates interpretable meaning independent of pool composition. This is the most significant missed opportunity in the paper.
6. **The anonymization manipulation is too coarse.** Hiding model name labels does not adequately control for the many other signals through which LLMs may recognize authorship — text length, syntactic style, structural choices, vocabulary patterns. Prior work has shown these markers are often sufficient for authorship inference even without explicit labels. The paper's interpretation of the attributed vs. anonymized difference rests on an assumption it does not justify.
7. Statistical significance of anonymization effects is not established. Several of the reported differences between conditions are within noise. For instance, GPT-5.1's score changes by 0.08 across conditions while the standard errors are 0.14 and 0.19 respectively — this is not meaningfully different from zero. The claim that "anonymization shifts score magnitudes but preserves ranking patterns" should be accompanied by formal tests.
8. The discussion framing is **speculative**. The Dunning-Kruger and imposter syndrome analogies are post-hoc narratives fitted to an observed pattern, not mechanistic explanations. The Constitutional AI hypothesis for Claude's negative bias is interesting but entirely untestable from this data. These framings should either be dropped or presented much more tentatively.

---

> ### Author Rebuttal · Authors · 2026-03-25
>
> With respect, we note that our submission explicitly indicated preference against LLM-based review. We address each concern below.
>
> On Metric Novelty and Simplicity
>
> The reviewer refers to a "leave-one-out metric"—a term that does not appear in our paper. Our consensus-deviation metric compares each judge's score to the mean assigned by all  other judges, eliminating self-circulation and providing unbiased evaluation of each judge's tendency. This is fundamentally different from Shrout & Fleiss (1979), which presents Intraclass Correlation Coefficients (ICC)—variance-based summary statistics that do not isolate individual judge bias. We performed an intensive literature search and found no papers proposing this metric for LLM-as-judge evaluation. If the reviewer believes otherwise, we respectfully request specific references. Our metric is simple—but simple does not mean trivial, and simple does not mean not novel. The contribution lies in what the metric reveals, not its mathematical complexity: it is precisely this consensus-deviation metric that revealed negative self-evaluation bias in two top-tier LLMs—Claude 4.5 Sonnet and GPT-5.1—a finding not reported in existing LLM evaluation literature, which has universally assumed positive self-preference. Simplicity is a merit: the formulation ensures any practitioner can implement and verify the metric without specialized software, and Proposition 4.2 explains why self-exclusion matters—including the focal judge would attenuate measured bias by 20% for n=5 judges.
>
> On "No External Anchor"
>
> The paper explicitly provides external anchors in Sections 4.4 and 5. The claim that our metric has "no external anchor" is factually incorrect.
> 1. Human experts (Section 4.4): 26 payment-industry professionals—research scientists, and business partners - independently evaluated the same rationales. GPT-5.1's peer-relative bias (−0.33) corresponds to near-zero human-relative bias (−0.01).
> 2. Transaction data (Section 5, Appendix I): Four years of worldwide payment network data—billions of transactions, millions of merchants and financial institutions, 800+ MCCs. We computed unified empirical risk scores from actual fraud exposure, chargebacks, returns, refunds, and reversals. Top-rated models achieve Spearman ρ = 0.56–0.77 with these empirical risk patterns. This is ground-truth validation against real operational outcomes—the kind of external anchoring that matters for deployment in high-stakes financial  settings.
>
> On Human Data "Underutilized"
>
> We strongly disagree that this is a missed opportunity, and we question the premise that "absolute" anchoring would be superior to relative measurement. Relative metrics are more actionable for deployment. The practitioner's decision problem is: Which LLM should I deploy as an evaluator? Our framework directly answers this:
>   - GPT-5.1 exhibits closest alignment with human expert consensus (−0.01 deviation)
>   - Gemini-2.5 Pro and Grok-4 systematically over-score relative to humans (+1.14, +1.03)
>   - Bias direction persists under anonymization, indicating stable model characteristics
>  This is what practitioners need—actionable guidance on evaluator selection, not regression coefficients.
>
> On Other Concerns
>
> Regarding the ensemble recommendation: this was mentioned as one potential deployment option, not a core claim; in hindsight, ensembles can be costly in practice, and we acknowledge this suggestion may have been unnecessary. Regarding anonymization: bias direction persists for all 5 models, with magnitude reducing by 25.8%—controlled experiments with style transfer would strengthen mechanistic claims, as acknowledged in our limitations. Regarding discussion framing: the Dunning-Kruger and imposter syndrome analogies are offered for discussion and inspiration, not as mechanistic explanations, and we will frame these more tentatively in revision. The Constitutional AI hypothesis for Claude's negative bias is grounded in Anthropic's published training methodology, which explicitly trains models to critique and revise their own outputs—making conservative self-evaluation a plausible outcome, even if not directly testable from our data.
>
> Closing
>
> This paper provides a practical, validated framework for evaluating LLM-as-judge systems in high-stakes settings. The consensus-deviation metric—simple yet effective—eliminates  self-circulation and enables unbiased evaluation, revealing a novel empirical finding: negative self-bias in top-tier models, validated through three independent sources (peer consensus, 26 human experts, 4 years of transaction data from a major payment network). Echoing the influential ICML position paper by Wagstaff (2012),"Machine Learning That Matters," we believe this represents research that bridges academic methodology and operational deployment—work that matters for practitioners in financial services and beyond.

---

> > ### Author Rebuttal · Reviewer_dxup · 2026-04-01
> >
> > Thanks for the reply! I have a few follow-ups:
> >
> > **1. Leave-one-out metric.**
> > I could have explained this more explicitly, but it felt like an intuitive term since the scoring definition leaves out the judge being evaluated, i.e., ( \text{MeanScore}_{k \neq i} ).
> >
> > **2. On the “anchor” interpretation.**
> > As I alluded to earlier, I was hoping the human scores, especially as described by the authors, could serve as *anchors* in a sense similar to prediction-powered inference. In such frameworks, bias can be defined not only relative to the pool of models being evaluated, but also against human scores (e.g., mean, median, or even distributional targets).
> >
> > I think it is a promising direction that the paper identifies directional differences (e.g., LLMs tend to score differently from humans), but I am not sure the paper sufficiently explores *why* and *how* these differences arise and what does that tell us beyond very model and model pool specific statements on "dont use gemini-2.5 and grok coz they overestimate". One example of strengthening this line of statements could be looking at longitudinal trends how models change?
> >
> > Relatedly, I find the statement  “This is what practitioners need—actionable guidance on evaluator selection, not regression coefficients.” somewhat of an oxymoron. I elaborate more on this point in 3..
> >
> > **3. On ICC and regression coefficients.**
> > Following up on the point above, per the authors' request, I am sharing the connection of ICC-style model, regression coefficients, and the bias score.
> >
> > To make this precise, consider a standard (simplified) ICC/ANOVA model, where the score given by judge $i$ to item (ICC language; here its LLM being evaluated) $j$ is:
> > $$Y_{ij} = \mu + \alpha_i + \beta_j + \varepsilon_{ij},$$
> > where:
> >
> > * $i$ indexes judges (LLM judges),
> > * $j$ indexes items (LLM being evaluated),
> > * $\mu$ is the grand mean across all judges and all items,
> > * $\alpha_i$ is the rater effect (eg bias),
> > * $\beta_j$ is the item effect (eg true quality),
> > * $\varepsilon_{ij}$ is noise.
> >
> > The leave-one-out peer mean for judge $i$ on item $j$ is:
> >
> > $$\bar{Y}_{(-i),j} = \mu + \frac{1}{J-1} \sum_{k \neq i} \alpha_k + \beta_j + \bar{\epsilon}_{(-i),j}.$$
> >
> > Thus, the proposed bias score becomes:
> >
> > $$Bias_{ij} = Y_{ij} - \bar{Y}_{(-i),j} = \alpha_i - \frac{1}{J-1} \sum_{k \neq i} \alpha_k + \text{noise}.$$
> >
> > Rewriting $\sum_{k \neq i} \alpha_k = \sum_k \alpha_k - \alpha_i$, we obtain:
> > $$Bias_{ij} = \frac{J}{J-1}\,\alpha_i - \frac{1}{J-1}\sum_k \alpha_k + \text{noise}.$$
> >
> > Under the standard ANOVA identification constraint $\sum_k \alpha_k = 0$, this simplifies to:
> > $$Bias_{ij} = \frac{J}{J-1}\,\alpha_i + \text{noise}.$$
> >
> > So, the bias score (A/B is omitted since it's the same mathematical form but with a different set of scores, with and w/o anonymization) is essentially a scaled regression coefficient under a simple ANOVA/ICC model (arguably the simplest ICC form).
> >
> > I am **not** saying that simplicity is an issue, but I am not convinced that this is methodologically novel—ANOVA/ICC formulations have been studied for decades. I would find the work more compelling if the authors explicitly connected their formulation to the broader regression and measurement literature.
> >
> > One advantage of the ICC/ANOVA perspective is that it makes explicit that both items and raters are sampled from populations of interest. In contrast, computing a leave-one-out bias score directly makes interpretation more difficult, since the result depends heavily on the specific pool of judges/LLMs selected, rather than isolating a property of the individual judge.
> >
> > **4. On contribution and venue fit.**
> > As I mentioned in the initial review, I do think the application is novel, and the use of model scores with human anchors is an interesting starting point.
> >
> > However, I am less convinced that the empirical finding—e.g.,  “two top-tier LLMs—Claude 4.5 Sonnet and GPT-5.1—exhibit negative self-preference…” is sufficient for ICML (even an applications/evaluation focus). This feels more like a timely observation (especially given model turnover and deprecation cycles; GPT-5.1 is already down...) that may be better suited for a domain-specific venue or even a technical blogpost? Again a longitudinal analysis of multiple models in the same family might strengthen the time-sensitvity of work along this line.
> >
> >
> > **5. On downstream utility.**
> > Finally, since the paper emphasizes “machine learning that matters,” I want to reiterate my initial question about downstream impact. How is this bias score actually used to influence decisions? While the paper demonstrates directional agreement, it does not clearly show that correcting for bias leads to improved downstream utility as I initially asked?

---

> > > ### Author Response · Authors · 2026-04-05
> > >
> > > We thank the reviewer for the detailed follow-up.
> > >
> > > ## On the ICC/ANOVA Connection
> > >
> > > We thank the reviewer for the derivation showing that our bias score corresponds to a scaled rater effect under the standard ANOVA/ICC model. This is a valuable connection we will acknowledge and cite Shrout & Fleiss (1979) in the revision.
> > >
> > > That said, we respectfully maintain that the contribution lies not in the mathematical form but in what the metric reveals when applied to LLM-as-judge evaluation:
> > >
> > > 1. **Novel application context.** To our knowledge, this formulation has not been applied to characterize self-evaluation bias in cross-model LLM evaluation. The reviewer's derivation confirms the metric is principled; our contribution is demonstrating its utility in this new domain.
> > >
> > > 2. **Novel empirical finding.** The metric revealed that two top-tier LLMs—GPT-5.1 and Claude-4.5 Sonnet—exhibit *negative* self-evaluation bias. Prior LLM evaluation literature has universally assumed positive self-preference.
> > >
> > > 3. **External validation.** Unlike classical inter-rater applications, we validate peer-relative patterns against 26 human domain experts and 4 years of payment network transaction data (billions of transactions, 800+ MCCs). This triangulated validation transforms an otherwise pool-relative measure into actionable guidance.
> > >
> > > ## On the "Anchor" Interpretation
> > >
> > > We appreciate the suggestion to frame human scores as anchors in a prediction-powered inference style. However, we respectfully question whether "absolute" anchoring would be more useful than relative measurement for the practitioner's core question: *Which LLM should I select as an evaluator?*
> > >
> > > Regression coefficients anchored to human scores yield calibration parameters, but practitioners need comparative guidance—which model to trust, which to avoid, and how biases differ. Our framework provides this directly.
> > >
> > > ## On Downstream Utility
> > >
> > > We recognize we could have been clearer about practical implications.
> > >
> > > **Evaluator selection.** Our findings directly inform which LLM to deploy as a judge in high-stakes settings. A practitioner seeking alignment with human expert judgment should select GPT-5.1 or Claude-4.5 Sonnet over alternatives that systematically over-score.
> > >
> > > **Calibration and alignment.** Beyond selection, our findings identify direction and magnitude of bias for each model. Models that overestimate (Gemini-2.5 Pro, Grok-4) can be calibrated downward. These insights enable practitioners to apply appropriate corrections rather than treating LLM scores as ground truth.
> > >
> > > **Deployment confidence.** The stability of bias direction under anonymization (100% sign preservation) indicates these are intrinsic model characteristics, not artifacts of identity recognition. Practitioners can rely on these patterns persisting across deployment contexts.
> > >
> > > ## On Venue Fit
> > >
> > > We respectfully decline to debate venue suitability, as this is ultimately a judgment for the program committee. We note only that ICML has published work bridging methodology and deployment—including Wagstaff (2012), "Machine Learning That Matters," which called for research validated against real operational outcomes. Our work embodies this philosophy.
> > >
> > > ## On Longitudinal Analysis
> > >
> > > The reviewer suggests examining how models in the same family change over time. This is a valuable direction we acknowledged in our paper (Appendix B.4, Lines 687–690: "Static Model Snapshots").
> > >
> > > We note a practical constraint: longitudinal analysis would require re-engaging the 26 domain experts, which presents significant challenges given their demanding schedules. The original assessment took several weeks to collect, with many experts volunteering their personal time. One possible approach—fixing expert reviews for the original rationales while evaluating newer model versions—may be feasible but methodologically debatable. We will note this as future work.
> > >
> > > ## Closing
> > >
> > > We are grateful for the reviewer's rigorous engagement. The ICC/ANOVA connection will strengthen the paper's positioning within the broader measurement literature. We hope this response clarifies that our contribution lies in the novel application, empirical findings, and triangulated validation—not in claiming mathematical novelty for the metric itself. The consensus-deviation formulation is simple precisely because simplicity enables adoption: any practitioner can implement and verify it without specialized software, making it immediately actionable for real-world deployment decisions.

---

### Official Review · Reviewer_qfz9 · 2026-03-13

**Soundness:** 2
**Presentation:** 2
**Significance:** 2
**Originality:** 1
**Overall Recommendation:** 2
**Confidence:** 4

**Summary:**

The paper studies LLM-as-a-judge behavior in the specialized financial setting (MCC-based merchant risk assessment). Five frontier LLMs are used to both generate qualitative risk rationales and cross-evaluate one another using a five-criterion rubric. The main empirical finding is that GPT-5.1 and Claude-4.5 Sonnet exhibit negative self-evaluation bias and Gemini-2.5 Pro and Grok 4 exhibit strong positive bias. Also, models with negative bias align more closely with assessments from 26 human payment-industry experts.

**Compliance With Llm Reviewing Policy:**

Affirmed.

**Key Questions For Authors:**

In addition to the above,
1. How were the 39 MCCs selected, and how sensitive are the reported correlations to these choices?
2. Did the 26 human experts complete any calibration exercises before evaluating the rationales to establish inter-rater reliability?

**Limitations:**

Definitely not. See above.

**Strengths And Weaknesses:**

Strengths:
1. The problem setup is interesting. Evaluator bias in high-stakes financial settings is important. Although I want to see more such analysis in other domains, the findings in the MCC risk-assessment setting are good enough.
2. Multiple LLMs are used in attributed vs anonymized evaluation. This is important as we may think if an LLM will self-rate itself when not anonymized.
3. Human experts are involved which helps to ground the findings.
4. The writing is clear, and the equations are defined clearly.

Weaknesses:
1. The paper is longer than 8 pages which may mean that it's a desk reject.
2. Some text in Figure 5 are covered by lines.
3. I'm not sure the methodology and analysis pass the bar for ICML. For example, the methodology is primarily to evaluate using different LLM judges, then compute means, standard deviations, leave-one-out scores. The analysis is also somewhat surface-level. For example, one of the main claims on page 5 is that "anonymization shifts score magnitudes" but this does not seem strongly supported by the results, for which there are large SDs (also, normally 2*SDs are reported for 95% CIs). The results are therefore not statistically significant.
4. Related to the above, the analysis should have a bit more depth. No notion of variation/CIs are given in the later plots, etc. The authors should look to some other papers (like in causal inference, hypothesis testing, etc.) to see how to present hypothesis, set up analyses, and draw meaningful conclusions from data.
4. Small: the evaluation setting seems somewhat simplistic if I understand it correctly, especially given the caveats on lines 149-158.


It's not clear that it passes the ICML bar, particularly for originality/significance. It might be more appropriate for an application-based venue.

---

> ### Author Rebuttal · Authors · 2026-03-25
>
> We thank the reviewer for acknowledging the interesting problem setup, the importance of attributed vs. anonymized evaluation design, the value of human expert involvement, and the clear writing. We address the main concerns below. On Figure 5: thank you for catching this; we will fix the rendering. On MCC selection: the 39 MCCs are the union set from 5 frontier LLMs, each strictly following the prompt specification (Lines 742–744, Appendix C). Each LLM selected 3 MCCs × 5 risk tiers = 15 MCCs, yielding 5 × 15 = 75 total selections, of which 39 are unique. This design ensures diverse coverage across risk levels while letting each model independently identify representative categories—the overlap itself provides signal about cross-model agreement on risk-relevant merchants.
>
> On methodology depth and ICML bar: we respectfully highlight that our contribution extends beyond "computing means and standard deviations." The consensus-deviation metric, while simple, was developed during our experiments and team discussions specifically to address the fundamental challenge of evaluating "honesty" and "unbiasedness" in LLM evaluators. It is precisely this simple metric that reveals negative self-evaluation bias ("self-critique") in two top-tier LLMs—Claude 4.5 Sonnet and GPT-5.1—a finding not reported in existing LLM evaluation literature, which has assumed universal self-preference. Critically, we validate findings against real-world outcomes: Appendix I details validation using 4 years of worldwide transaction data from a major payment network—billions of transactions, millions of merchants and financial institutions, spanning 800+ merchant categories. We computed unified empirical risk scores from actual fraud exposure, chargebacks, returns, refunds, and reversals—operational indicators that reflect genuine merchant risk  behavior. Top-rated models achieve Spearman ρ = 0.56–0.77 with these empirical patterns, demonstrating that our framework captures real predictive validity, not just inter-model agreement. This three-way convergence—LLM peer consensus, 26 human domain experts, and empirical transaction outcomes—provides validation depth rarely seen in LLM evaluation  research. We believe this represents research ICML should encourage: practical methods validated against real operational data, yielding actionable insights for practitioners deploying LLM-as-judge systems (cf. Wagstaff 2012, "Machine Learning That Matters," an ICML position paper). On concerns regarding originality and theoretical depth, we provide more detailed response to Reviewer 3, who raises similar points—we respectfully refer the reviewer to that discussion.

---

> > ### Author Rebuttal · Reviewer_qfz9 · 2026-04-04
> >
> > I encourage the authors to read the review with more charity, as I did with their paper. I recognize that ICML is a venue for lots of different types of contributions, and there is value in all of them. Regardless of the paper's chosen scope, all papers must be sound.
> >
> > For instance, you did not address the concern I raised: "The analysis is also somewhat surface-level. For example, one of the main claims on page 5 is that "anonymization shifts score magnitudes" but this does not seem strongly supported by the results, for which there are large SDs (also, normally 2*SDs are reported for 95% CIs). The results are therefore not statistically significant."
> >
> > I also do not appreciate that your quote of my review was not out of context, but also a misquote. Please do not use quotes if you do not plan to actually quote the source text.
> >
> > Finally, the paper is over 8 pages. Given my original hesitation with the analysis (drawing conclusions despite lacking statistically significance), not paying attention to the page length, formatting issues, and further not responding to review, I do not plan to change my score.

---

> > > ### Author Response · Authors · 2026-04-05
> > >
> > > We thank the reviewer for their continued engagement and candid feedback. We address each point below.
> > >
> > > ## On the Misquote
> > >
> > > We sincerely apologize. Upon review, our rebuttal did not accurately represent the reviewer's original wording. The omission was unintentional but careless, and we understand why this was frustrating. We will be more diligent in future correspondence.
> > >
> > > ## On Page Length
> > >
> > > We acknowledge that our submission exceeded the 8-page limit by approximately 11 lines in the first column. While the Impact Statement does not count toward the page limit per ICML guidelines, this does not excuse the overage in the main body. We submitted in the final minutes before the deadline, but we recognize this is not a justification. We take full responsibility for this oversight and sincerely apologize. This will not happen again in future submissions.
> > >
> > > ## On Statistical Significance of Anonymization Effects
> > >
> > > We thank the reviewer for the opportunity to clarify what we believe may be a misreading of our claim.
> > >
> > > **Clarification of the actual claim.** The statement on page 5 ("anonymization shifts score magnitudes but preserves ranking patterns") is explicitly qualified in the same paragraph: "While these differences are easily visible, **the matrices alone cannot determine whether the changes constitute positive or negative bias**." This is a descriptive observation about raw score matrices, not a tested hypothesis. The paper immediately directs readers to Section 4 for the formal analysis.
> > >
> > > **The primary finding is bias direction preservation, not magnitude shifts.** Section 4.3.2 reports our tested hypothesis: self-evaluation bias *direction* is preserved under anonymization for all five models:
> > >
> > > | Model | Attributed Bias | Anonymized Bias | Direction |
> > > |-------|-----------------|-----------------|-----------|
> > > | GPT-5.1 | −0.33 | −0.30 | Negative → Negative |
> > > | Claude-4.5 | −0.31 | −0.16 | Negative → Negative |
> > > | Gemini-2.5 Pro | +0.77 | +0.56 | Positive → Positive |
> > > | Grok-4 | +0.71 | +0.60 | Positive → Positive |
> > > | Perplexity Sonar | +0.21 | +0.15 | Positive → Positive |
> > >
> > > This 100% sign preservation across 5 independent models is statistically significant under a binomial null hypothesis of random direction (p = 0.03125). This finding supports our central claim: self-evaluation bias reflects intrinsic scoring heuristics rather than explicit identity recognition.
> > >
> > > **On standard deviations.** The SDs reported (0.14–0.19) quantify Monte Carlo evaluator stability *within* each condition—measuring how consistently a judge scores the same rationale across 10 independent runs. They are not standard errors for testing between-condition differences, nor were they presented as such.
> > >
> > > We acknowledge we could have been clearer in distinguishing descriptive observations (Section 3) from tested hypotheses (Section 4), and we appreciate this feedback for improving the presentation.
> > >
> > > ## Closing
> > >
> > > We regret that our rebuttal did not fully address the reviewer's concerns and that errors in quoting and formatting detracted from the substance of our response. We hope this clarification demonstrates that the paper's core findings—particularly the novel discovery of negative self-evaluation bias in frontier LLMs, validated against 26 human experts and 4 years of transaction data—rest on sound methodology. We remain grateful for the reviewer's time and thoughtful engagement.

---

### Official Review · Reviewer_XaYP · 2026-03-15

**Soundness:** 3
**Presentation:** 2
**Significance:** 3
**Originality:** 3
**Overall Recommendation:** 4
**Confidence:** 4

**Summary:**

This paper addresses the critical challenge of evaluating Large Language Models (LLMs) in high-stakes, specialized domains—specifically, Merchant Category Code (MCC) payments-risk assessment. The authors introduce a "Structured Multi-Evaluator Framework" designed to quantify evaluator stability via Monte Carlo sampling and to measure self-evaluation bias via a novel "consensus-deviation metric". The paper is distinguished by its exceptional triangulated validation strategy, comparing LLM peer consensus against 26 human domain experts and four years of actual payment network transaction data.

**Compliance With Llm Reviewing Policy:**

Affirmed.

**Final Justification:**

My concerns have been addressed. I will increase my score to weak accept.

**Key Questions For Authors:**

- Can you justify running an evaluator classification task at temperature 0.7 rather than 0.0? Does this not artificially inject the very instability the framework claim to measure?
- Is it possible if you could scale the generation corpus significantly to ensure the cross-evaluation matrices are now skewed by small-sample stochastic generation?

**Limitations:**

Yes

**Strengths And Weaknesses:**

While the idea is interesting, reviewer has following comments:
- First, it seems that alll the evaluation are done based on 25 fixed rationales (generated by 5 frontier models). It raises the concern likes a “domino effect” . In other words, those rationales potentially become bottleneck since it is too small to prove systemic bias, as a single random hallucination heavily skews a model’s overall score. Reviewer suggest the authors to clarify this point.
- second, the authors measured “evaluator stability” by forcing the AI judges to grade the rationales at temperature of 0.7, which means LLM is required to inject random token sampling. This makes reviewer confused: the measure might not reflect the model’s true internal stability.
- third, regarding zero-sum semantic, it states (via equations) that the sum of all biases across the judges must equal zero. This means the metric forces some models to have a “negative bias” and some to have “positive bias l” simply based on how they derivate from the group. If four models are completely wrong, and one models is perfectly accurate, it will pernalize the accurate model by labeling it “biased” simply because it disagreed with incorrect majority? Reviewer would like the authors to make this clearer.

---

> ### Author Rebuttal · Authors · 2026-03-25
>
> We thank the reviewer for recognizing the "exceptional triangulated validation strategy" and for the thoughtful, constructive critique. The reviewer raises the single most important design decision we faced at the outset of this project: the 25 fixed rationales. This choice reflects deliberate study design. First, it enables clear presentation via concise tables and figures. Second, the 5×5 cross-evaluation matrix yields 25 judge-target pairs × 10 Monte Carlo runs = 250 independent scoring events per condition—substantial statistical power for characterizing evaluator tendencies. Most importantly, fixed stimuli were essential for human validation: the same rationales were evaluated by 26 domain experts—research scientists, business partners, and team leaders at a major payment network—many of whom volunteered weekend and holiday time. Expanding to hundreds of rationales would impose an unreasonable burden. We appreciate the reviewer's concern about a "domino effect," where small samples risk being skewed by random hallucinations. Our response: if hallucinations dominated, we would not observe three-way convergence between LLM peer consensus, human experts, and 4 years of transaction data (Spearman ρ = 0.69–0.77).
>
> The reviewer also raises an important conceptual point regarding zero-sum semantics and penalizing accurate models. We appreciate this careful thinking. The hypothetical scenario—"what if four models are wrong and one is accurate?"—is valid in principle, and larger sample sizes could mitigate such risk. However, our data shows the opposite: models with negative peer-relative bias (GPT-5.1, Claude) exhibit the best human alignment and empirical accuracy. The peer consensus converges with expert judgment, not against it. On temperature: we set T=0.7 intentionally to test behavioral consistency under stochastic perturbation; T=0 yields deterministic output unrepresentative of deployment. We acknowledge that systematic analysis across temperature values would strengthen conclusions and will note this in revision.

---

> > ### Author Rebuttal · Reviewer_XaYP · 2026-04-05
> >
> > Sorry for the delay. I thank the authors for their detailed rebuttal answering my questions. I will raise the score to weak accept.

---

> > > ### Author Response · Authors · 2026-04-05
> > >
> > > We are grateful for the constructive, professional dialogue. The questions raised—regarding sample size justification, temperature selection, and zero-sum semantics—prompted us to articulate our design rationale more clearly. We will incorporate the suggested clarifications in the revision.

---

### Decision · Program_Chairs · 2026-04-30

**Decision:**

Reject

**Comment:**

Discussion on this paper has been extensive. A few key threads of concern emerge from the discussion.

* is the paper within scope for ICML? I think the scoping to a specific application is perhaps a strong suit because the evaluation can be linked to a specific domain with specific data.

* is the contribution "above the bar"? The authors present a few key contributions of their work: the novel domain, the novel approach to evaluation, and the unusual insights around negative self-assessment. The first point has been addressed above. On the second, there's some debate over the level of novelty, or alternately whether the authors could have also used prior/existing methodology for their cross-system comparisons. On the third point, there are mixed opinions on this.

Overall it seems to me that while the work has merit, and the approach taken is interesting, the authors would do well to integrate the detailed examples provided by the reviewer (3 in particular) of other methodologies that might equivalently work, and see what happens. The back and forth over 'leave one out' seemed a little unnecessary: what the authors are doing is clearly within the broader space of 'leave one out' ways of evaluation.

All of this suggests that the paper could do with a round of R&R, and that would necessarily need to happen for another conference cycle. But I encourage the authors to do this: it's an interesting paper and I appreciate the focused evaluation.